



# Wind tunnel testing of wake steering with dynamic wind direction changes

Filippo Campagnolo[1], Robin Weber[1], Johannes Schreiber[1], and Carlo L. Bottasso[1]

[1]Wind Energy Institute, Technische Universität München, Garching bei München, D-85748 Germany,

*Correspondence to:* C.L. Bottasso (carlo.bottasso@tum.de)

**Abstract.**

The performance of an open-loop wake steering controller is investigated with a new unique set of wind tunnel experiments. A cluster of three scaled wind turbines, placed on a large turntable, is exposed to a turbulent inflow and dynamically changing wind directions, resulting in dynamically varying wake interactions. The changes in wind direction were sourced and scaled from a field-measured time history and mirrored on to the movement of the turntable.

Exploiting the known, repeatable and controllable conditions of the wind tunnel, this study investigates the following effects: fidelity of the model used for synthesizing the controller, assumption of steady state vs. dynamic plant behavior, wind direction uncertainty, the robustness of the formulation in regard to this uncertainty, and a finite yaw rate. The results were analyzed for power production of the cluster, fatigue loads and yaw actuator duty cycle.

The study highlights the importance of using a robust formulation and plant flow models of appropriate fidelity, and the existence of possible margins for improvement by the use of dynamic controllers.

## 1 Introduction

Wakes produced by upstream wind turbines have a profound influence on the performance of downstream machines. Compared to clean isolated conditions, waked turbines produce less power, approaching a 50% reduction for full-wake interaction (Mechali et al., 2006), and experience increased loading (Madjidian et al., 2011; Bustamante et al., 2015; Vera-Tudela and Kühn, 2017). The impact both in terms of lost production and increased loading is significant and has cascading effects on operation and maintenance (O&M) and lifetime. Probably one of the most direct indications of the impact of wakes outside of the scientific literature is given by the press announcement issued by Ørsted (formerly DONG) in October 2019. In this announcement, Ørsted, the largest offshore wind energy developer in the world, warned investors that it will not be able to meet its long-term financial targets. Next to market issues, "[...] the negative impact of two effects across our asset portfolio, i.e. the blockage effect and the wake effect" was listed as the main reason. In addition, Ørsted stated that "[...] underestimation of blockage and wake effects is likely to be an industry-wide issue" (Ørsted, 2019).

Wind farm control is widely recognized as one of the main solutions to mitigate wake effects (Gebraad et al., 2016; Fleming et al., 2016; Vali et al., 2017; Fleming et al., 2017; Raach et al., 2018; Fleming et al., 2019). In wind farm control, the turbines in a wind farm operate in coordinated, collaborative fashion. This stands in direct contrast to the standard, greedy approach





in which each machine works independently from the others to maximize its own power output —even if this is detrimental to the output of its neighboring turbines. A number of wind farm control strategies are currently being investigated, including static and dynamic induction control (Frederik et al., 2020), and wake steering (Knudsen et al., 2015; Fleming et al., 2016). Among these, wake steering is probably the most promising technique for practical field deployment and reports of field tests

have already been published (Fleming et al., 2017, 2019; Howland et al., 2019). This control technology is also offered as a feature for offshore wind farms by one of the leading wind turbine manufacturers (Siemens Gamesa, 2019).

Although a field demonstration is clearly the final litmus test for any technology, simulations with high fidelity models and scaled testing in wind tunnels offer some unique opportunities to improve knowledge and understanding. Bottasso et al. (2014) pioneered wind tunnel testing beyond pure aerodynamic investigations by developing several experimental applications based

on actively controlled scaled wind turbines. Campagnolo et al. (2016c) followed up with an experimental demonstration of closed-loop wake steering. In addition to their own scientific advances, these works provided comprehensive opportunities for the validation of simulation models (Wang et al., 2019).

The present paper follows in these same tracks. Here three scaled turbines are tested in a large boundary layer wind tunnel, where dynamic wind direction changes are generated by using a turntable. The three machines are governed by an open-loop

wake steering controller, while each machine is operated by its own closed-loop yaw, pitch and torque controller. The known, repeatable and controllable environment of the wind tunnel offers the opportunity to address some key questions:

– What are the effects of neglecting the dynamics of wake interaction by using a steady-state controller? And what are the additional effects caused by a limited yaw rate and a finite sampling time of the controller?

– What are the effects on performance of the fidelity of the underlying model used for control synthesis? Does it pay off to

20       use a better model, and what are the margins for improvement? Are conclusions different when looking at power, loads or actuation effort?

– What are the benefits of using a formulation that is robust in the face of uncertainties, as opposed to a naive deterministic approach? And is there a minimum wake interaction threshold below which it might be better not to use a wake steering controller?

This study is an initial effort to try and answer these questions.

The paper is organized as follows. Section 2 describes the experimental setup, including the scaled turbines, the tunnel sheared and turbulent inflow, and the generation with a turntable of dynamic wind direction changes that mimic actual field measurements. Since the ground truth wind direction is known in the case of the experiment, a filtering approach is described to provide the controller with a tunable level of uncertainty, with the goal of characterising its effects on performance. Section 3

describes the control formulation and implementation. A model-based robust formulation is used here, which first derives look-up table (LUTs) by an offline optimization and then interpolates within the LUTs at run-time based on the detected operating conditions. To explore the effects of varying model fidelity, three different farm flow models are considered: the standard FLORIS model (Doekemeijer et al., 2018), which lacks important effects such as wake steering and non-uniform inflow; an





improved version of the same model based on the learning of correction terms from operational data, termed FLORIS-Augm (Schreiber et al., 2019); and a purely data-driven model that, based on the accurate measurements that are possible in the wind tunnel, can be considered as an exact steady-state representation of the experiment. Section 4 presents an analysis of the experimental results. The non-robust formulation is analyzed first in terms of the effects on performance of uncertainty

level, finite yaw rate, neglected dynamics, and model fidelity. Next, the robust formulation is considered and compared to the non-robust one, looking at the metrics of power, fatigue loading and actuator duty cycle. Section 5 concludes the work and provides some initial answers to the questions posed above.

## 2   Experimental setup

The experimental setup is shown in Fig. 1: a small cluster composed of three scaled wind turbines is installed on the 13 m

diameter turntable of the atmospheric test section of the wind tunnel of the Politecnico di Milano (Bottasso et al., 2014). The turntable can be rotated by the angle $\Phi$ to simulate different wind directions. This is achieved by first lifting the turntable with an air cushion by approximately 20 mm, and then rotating it by means of a friction wheel. A dedicated controller is used to track the user-prescribed rotation time history. An optical encoder with an accuracy of $\pm 0.1$ deg is used as feedback. The turntable was in the lifted position throughout the course of each experiment.

The three turbines are aligned in a row with a longitudinal spacing of five rotor diameters (5D), and they are termed WT1 (upstream), WT2 (center) and WT3 (downstream). The wind direction $\Phi$ is zero when the turbine row is parallel to the wind tunnel center-line. In this position, the row of turbines is located -1.5D to the left of the center-line when looking upstream. $\Phi$ is positive for a clockwise rotation of the turntable viewed from the top (cf. Fig. 1); this means that a positive $\Phi$ corresponds to the wind blowing from the left of the row of turbines when looking upstream.

Rotating the turntable does not exactly correspond to a change in wind direction with respect to fixed ground. In fact, the scaled turbines experience a translational movement proportional to the angular speed of the turntable and to their distance from the center of rotation. This generates an additional flow velocity relative to the rotor, on average equal to approximately 0.04 m/s. In turn, this creates a small extra local wind direction change, quantified to less than 0.5 deg for the current setup and testing conditions. Other differences with respect to a real wind direction change are caused by the slight horizontal shear

present in the wind tunnel flow. The translational movement exposes the turbines to different flow speeds as they move laterally in the tunnel during the turntable rotation. This effect is not negligible, but it can be accounted for if the horizontal shear is known, as discussed later.

### 2.1   Wind turbine model

Three identical G1 scaled models with a rotor diameter, hub height and rated rotor speed of 1.1 m, 0.825 m and 850 rpm,

respectively, were used in the experiments. The models, already used in previous research projects (Campagnolo et al., 2016a, c, b), are equipped with active pitch, torque and yaw control. Strain gauges measure loads on the shaft and at the tower base.



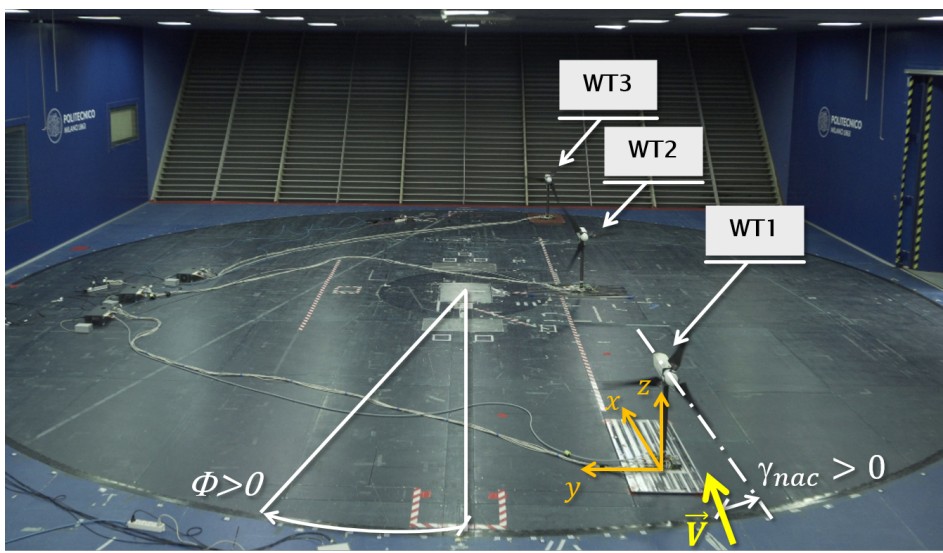

**Figure 1.** Experimental setup, showing the three model turbines mounted on the wind tunnel turntable. The $x$-$y$-$z$ frame is fixed with respect to the tunnel, and does not rotate with the turntable.

Further details about the G1 design, its aerodynamic performance and several of its applications can be found in Bottasso and Campagnolo (2019).

Each wind turbine is controlled with a dedicated real-time Bachmann M1 system, where supervisory control logic, pitch-torque-yaw control algorithms, and all necessary safety, calibration and data logging functions are implemented. Demanded
reference values for torque, pitch and yaw are computed by the wind turbine controller and then sent to the actuator control boards, where low level control functions are executed. The M1 system acquires torque, shaft bending moments and rotor azimuth position with a sample rate of 2.5 kHz, whereas all other measurements (tower base loads, blade pitch angles, wind speed and direction) are acquired with a sample rate of 250 Hz.

A standard power controller is implemented based on Bossanyi (2000), with two distinct control regions (Burton et al.,
2011). Below rated wind speed (region II), the blade pitch angle is held constant, while the generator torque is a quadratic function of the rotor speed that enforces a constant tip speed ratio (TSR). Above rated wind speed (region III), the generator torque is kept constant, while a proportional-integral (PI) controller changes the collective pitch of the blades to enforce a constant generated power.

The nacelle orientation $\gamma_{\mathrm{nac}}$ is positive for a counter-clockwise rotation when viewed from the top (cf. Fig. 1), and can be
varied at will with respect to the base. The positioning is achieved with a PI controller executed on the control board of the yaw motor. A yaw brake can be engaged once the nacelle reaches the desired position within a tolerance of $\pm 0.2$ deg. Whenever the reference orientation is changed, the brake and the motor are simultaneously actuated to ensure smooth transitions.



The wind farm controller was implemented on a desktop PC, communicating with the turbine controllers through the MOD-BUS protocol. This plant-level controller sets a desired misalignment angle $\gamma$ with respect to the wind for each turbine. A positive $\gamma$ corresponds to a counter-clockwise misalignment looking down onto the model, i.e. the opposite direction of $\Phi$. The relationship between wind direction, nacelle orientation and yaw misalignment angle is

$$\gamma = \gamma_{\mathrm{nac}} - \Phi. \tag{1}$$

Figure 2 shows the behavior of the G1 rotor thrust coefficient $C_{\mathrm{T}}$ and power coefficient $C_{\mathrm{P}}$ with respect to the rotor-effective wind speed $U_{\mathrm{REWS}}$ (top row of the figure) and with respect to the misalignment angle $\gamma$ (bottom row of the figure).

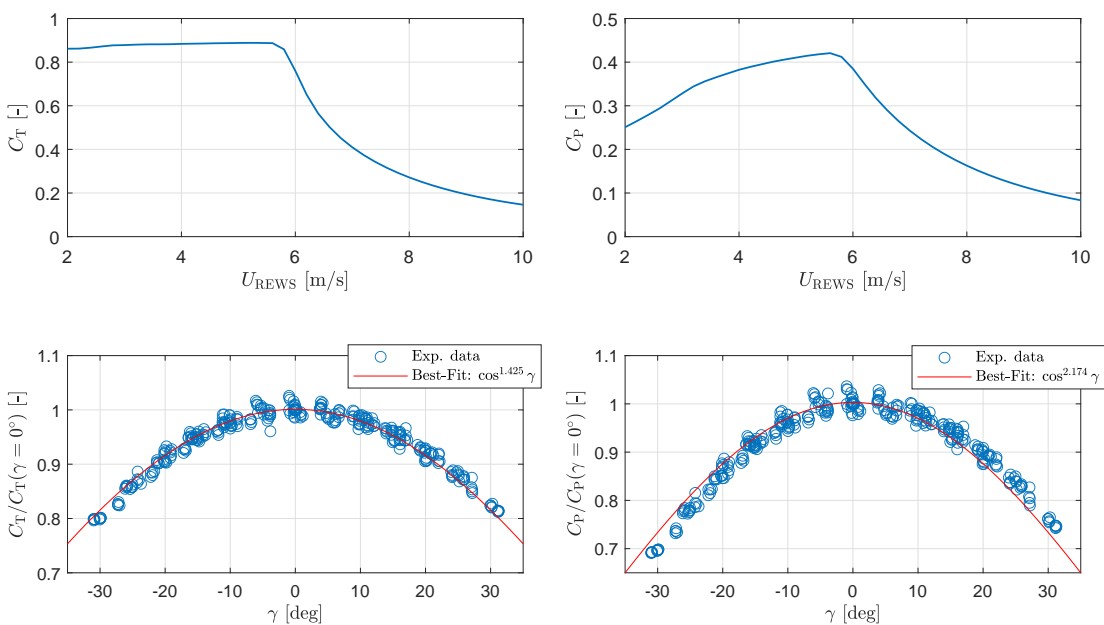

**Figure 2.** Top: $C_{\mathrm{T}}$ (left) and $C_{\mathrm{P}}$ (right) vs. rotor-equivalent wind speed $U_{\mathrm{REWS}}$. Bottom: $C_{\mathrm{T}}$ (left) and $C_{\mathrm{P}}$ (right) vs. misalignment angle $\gamma$.

The behavior of the thrust and power coefficients vs. rotor-effective wind speed was obtained by closed-loop simulations with FAST (Jonkman and Jonkman, 2018), with turbulent flow conditions similar to the ones generated in the wind tunnel in terms of speed and turbulence intensity. The blade aerodynamic model uses Reynolds-dependent airfoil polars tuned as described in Wang et al. (2020). The top right plot of Fig. 2 shows that the wind turbine $C_{\mathrm{P}}$ is affected by the Reynolds-dependency of its airfoil polars in region II (i.e. for wind speeds lower than approximately 5.7 m/s). At low winds, and hence at low rotational speeds, the blade airfoil efficiency is reduced because of the low chord-based Reynolds number, resulting in a reduction of $C_{\mathrm{P}}$. However, the Reynolds number has only a modest effect on the lift coefficient (Wang et al., 2020), thus resulting in an approximately constant $C_{\mathrm{T}}$ (cf. Fig. 2, top left).





The behavior of the thrust and power coefficients vs. misalignment angle was characterized with dedicated wind tunnel tests, conducted for $\gamma \in \pm 31$ deg with the turbine operating in region II. The results are reported in the bottom part of Fig. 2. The best-fitting cosine-law power-loss exponents equal $2.174$ and $1.425$ for the power and thrust coefficients, respectively.

## 2.2 Inflow characteristics

5 Spires placed at the inlet of the test section passively generate an atmospheric-like boundary layer. The flow was characterized with three-component constant-temperature hot-wire probes (CTA), scanning a vertical line 4D upwind of WT1. The vertical profiles of the longitudinal wind speed $U$ (normalized by the speed at hub height $z_H$) and the turbulence intensity (TI) are shown in the left and central plots of Fig. 3, respectively. The top and bottom points of the rotor are indicated with solid black lines, while a dashed black line indicates hub height. The vertical wind profile within the rotor disk is best-fitted by a power 10 law with exponent equal to $0.144$, while turbulence intensity at $z_H$ is approximately equal to $6\%$, mimicking typical neutral, offshore conditions (Hansen et al., 2012).

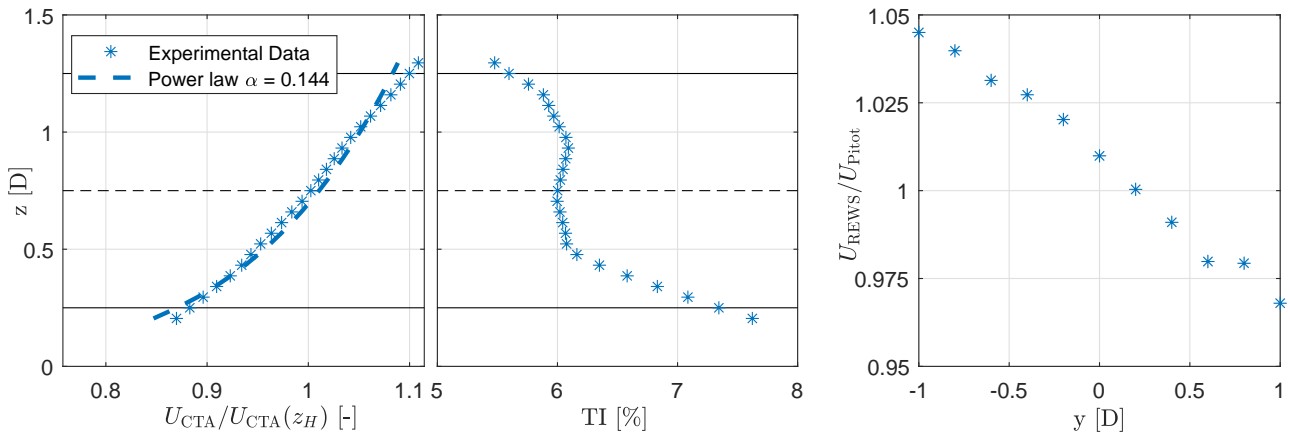

**Figure 3.** Characteristics of the wind tunnel boundary layer: vertical profiles of wind speed (left) and turbulence intensity (center), measured with CTA probes; lateral profile of the wind speed (right), measured using the rotor as a sensor.

For a correct interpretation of the wind farm control results, the small lateral non-uniformity of the wind tunnel flow needs to be taken into account (Wang et al., 2017). In fact, as the turntable is rotated, the turbines are also displaced laterally, thereby encountering slightly different ambient conditions. The ambient wind speed was measured by a pitot tube installed at hub 15 height, laterally shifted 1.5D to the left of the wind tunnel center-line and 3D upwind of WT1. The pitot tube is, therefore, in front of the turntable, and remains fixed with respect to the wind tunnel as the turntable is rotated. This means that the pitot tube is exactly in front of WT1 only for $\Phi = 0$, whereas it is laterally displaced with respect to the front turbine in all other cases. Hence, given the non-uniformity of the wind tunnel boundary layer, when the turntable is rotated the turbines are exposed to a local ambient flow that differs slightly from the one measured by the pitot tube.



To characterize this effect, one G1 was positioned at several different lateral locations $y$ across the wind tunnel (see Fig. 1). The local rotor-effective wind speed $U_{\mathrm{REWS}}$ was computed directly from the torque measured on the turbine for each location. The resulting lateral profile of the wind speed is reported in the right plot of Fig. 3. This diagram shows the presence of a horizontal shear with changes in wind speed up to $\pm 4\%$, for both left and right shifts with respect to the pitot tube. These

changes will clearly cause significant changes in power, due to its cubic dependency on speed.

### 2.3    Dynamic wind direction changes

Testing at scale not only implies different physical dimensions of the model, but also a scaling of time with respect to the original system. Specifically, the time speed-up factor is defined as $n_t = t_M/t_P$, where $t_M$ is the time of the scaled system and $t_P$ the time of the full-scale system (Bottasso et al., 2014; Canet et al., 2020). In this specific case, the G1 turbines represent

scaled models of an 8 MW full-scale machine with a rated rotor speed $\Omega_P = 10.5$ rpm (Desmond et al., 2016). Since the time speed-up factor can also be expressed as $n_t = \Omega_P/\Omega_M$, time flows faster by a factor of approximatively 80 in the wind tunnel than at full scale. Thus, one hour of testing in the tunnel corresponds to about 3.3 days in the field, an additional, valuable side effect of testing at scale.

   In order to obtain results that can be up-scaled, changes in wind direction simulated in the wind tunnel should realistically

mimic full-scale variations. To this end, a wind direction time history was measured at 1 Hz at an onshore test site located in northern Germany using a wind vane installed at a height of 89.4 m on a met-mast (Bromm et al., 2018). Within the available data set, five days of measurements were selected and scaled by $n_t$, obtaining a time history used for driving the turntable rotation. The data selection criteria were as follows:

   – Met-mast always fully out of the wakes of neighboring machines;

– Wind direction variations within the range $\pm 15$ deg as, given the experimental setup, wake interactions within the cluster are expected only for $\Phi \in \pm 15$ deg;

   – Enough data to draw statistically meaningful conclusions, using Fleming et al. (2019) as a guideline.

   The top plot of Fig. 4 reports the frequency spectrum of the scaled (i.e. sped-up) field-measured wind direction time series $\Phi_{\mathrm{Met}}$. The plot also shows the spectrum of the wind direction changes $\Phi_{\mathrm{CTA}}$ already naturally present (without using the

turntable) in the wind tunnel flow due to the generated turbulence, as measured with the CTA probes. The figure shows that there is a very good match at the high frequencies between the real flow and the one in the wind tunnel. On the other hand, it is also evident that the wind tunnel boundary layer misses completely the large amplitude fluctuations present in the field at scaled frequencies below about 0.66 Hz. Taking into account the time scaling factor, this means that wind direction fluctuations characterized by a period above approximatively 2 min are missing from the tunnel flow. Since these are the dominant wind

direction changes for wind farm control (Simley et al., 2019), a way is needed to fill the lower band of the spectrum.

   With the turntable, these missing low-frequency wind direction fluctuations can be filled in. Unfortunately, an exact reproduction of the complete spectrum is not possible due to hardware limitations. In fact, the rotational acceleration of the turntable



**Figure 4.** Top: spectrum of the sped-up field-measured wind direction time series $\Phi_{\mathrm{Met}}$ (solid blue), the turbulence induced wind direction changes in the tunnel $\Phi_{\mathrm{CTA}}$ (dashed red) and the turntable rotation $\Phi_{\mathrm{turn}}$ (dash-dotted orange). Center: spectrum of the sped-up wind direction changes in the field $\Phi_{\mathrm{Met}}$ (solid blue) and the combined wind direction changes in the wind tunnel $\Phi_{\mathrm{CTA}} + \Phi_{\mathrm{turn}}$ (dashed green). Bottom: time history of the 2-min average of the sped-up field-measured wind direction (blue), compared to the time history used to drive the turntable rotation (dash-dotted orange).





is limited by the maximum force that can be exerted with the driving friction wheel. At higher accelerations, inertial effects on the models would also have to be taken into account. To obtain a time series that could be followed by the turntable, piecewise cubic splines were used to best fit a 2-min moving-average of the wind direction time history, under the constraints of maximum achievable acceleration and velocity. The resulting time series $\Phi_{\mathrm{turn}}$ is compared to the sped-up 2-min average of $\Phi_{\mathrm{Met}}$

in the lower plot of Fig. 4.

The central plot of Fig. 4 shows the spectrum of the resulting wind directions obtained by combining the natural changes present in the wind tunnel flow with the artificial ones generated by the turntable. A comparison with the field-measured spectrum shows that the two match very well at the lowest and highest frequencies. On the other hand, the combined wind tunnel flow has a gap in the range $0.04 - 0.66$ Hz, which corresponds to directions changes between 2 and 30 min at full scale.

Filling this gap would require a modification to the actuation system of the turntable, which was unfortunately not possible within the scope of the present work.

## 3   Open-loop wake steering controller

The wind farm control strategy is the open-loop algorithm sketched in Fig. 5. The algorithm consists of a model-based optimization that produces a look-up table (LUT) of discrete set-points, followed by an interpolation within the pre-computed table

at given instantaneous ambient conditions.

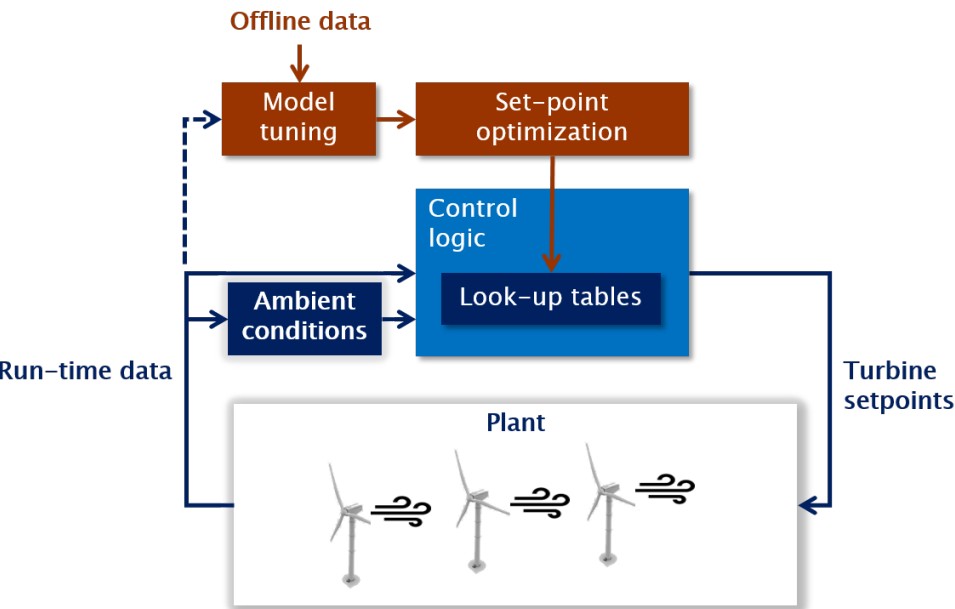

**Figure 5.** Wind farm control scheme. The option to update the model at runtime to recompile the LUTs (dashed line) was not used in the present work.





A wind farm flow model is first calibrated with the use of pre-existing data (and possibly re-tuned on-line during operation, although the present work did not make use of this possibility). Based on this model, an optimization is performed off-line to compute the optimal set-points of each machine in the farm that minimize a cost function for given ambient conditions. In this work, the set-points consist of yaw offsets of each turbine with respect to the ambient flow direction. To understand the effects of model fidelity on the controller performance, LUTs were computed based on the three different flow models described in §3.1.

During operation, filtered ambient wind conditions are computed, including wind direction, wind speed, and turbulence intensity (because of its effect on wake recovery). These conditions can be estimated from the operational data of the turbines (Schreiber et al., 2018), or simply by a met mast (Fleming et al., 2019). Based on the ambient wind conditions, the control logic interpolates within the LUT to compute the current set-points, which are then dispatched to each individual wind turbine. The process of ambient condition estimation, LUT interpolation and dispatching is repeated with a desired frequency.

Similar controllers have been recently implemented and tested in the field (Fleming et al., 2019). However, the implementation in a wind tunnel experiment has some specific features, which are discussed next.

The ambient conditions in the experiment are characterized by constant mean wind speed and turbulence intensity, but variable low frequency wind direction changes generated by the turntable. The top plot of Fig. 6 shows the combined wind direction time history $\Phi_{\mathrm{CTA}} + \Phi_{\mathrm{turn}}$, its 1.5 s moving average, and the turntable rotation $\Phi_{\mathrm{turn}}$. In the experiments, the true wind direction is therefore known through the turntable encoder with a high accuracy and signal-to-noise ratio, something that is hardly possible in the field.

The turntable signal is filtered and provided as wind direction input to the controller. By filtering this signal, the controller reacts only to low-frequency fluctuations and neglects higher frequency turbulent changes, which is desirable for yaw-based control (Simley et al., 2019; Fleming et al., 2019). However, increasing the filtering action generates longer delays, which has the effect of changing the wind direction seen by the controller with respect to the true one. This fact was exploited here to generate a variable level of uncertainty and study its effects on the controller performance. To assess the effects of filtering (i.e. uncertainty), three values of the moving average time window were considered and used as input for the controller, namely $T_{\mathrm{MAvg}}$ = 1.5, 7.5 and 15 s, which correspond to 2, 10 and 20 min at full scale. The effects of the filter on the wind direction time series are shown in the bottom plot of Fig. 6.

At runtime, the controller outputs the optimal yaw misalignment angle $\gamma_1$ for WT1 and $\gamma_2$ for WT2 at each time step (equal to 0.75 s, which corresponds to 1 min at full scale), whereas the downstream turbine WT3 adopts a standard wind-tracking yaw strategy with the same time step. To guarantee a more precise yaw misalignment (Bossanyi, 2018), a direct control of the nacelle orientation was preferred to the indirect approach used by Fleming et al. (2017, 2019). In this method, the required absolute nacelle orientation is computed from Eq. (1) as $\gamma_{\mathrm{nac}} = \gamma + \Phi_{\mathrm{meas}}$, where $\Phi_{\mathrm{meas}}$ is the measured wind direction (i.e. the filtered turntable encoder signal). The nacelle is then actuated with a maximum yaw rate $\dot{\gamma}_{\mathrm{max}} = 10$ deg/s (0.125 deg/s at full scale) to limit gyroscopic loads on the G1. As discussed later, the maximum yaw rate has a significant effect on performance; it should be noted that the value chosen here is lower than the 0.3 deg/s at full scale used in other publications (Bak et al., 2013; Jonkman et al., 2009).



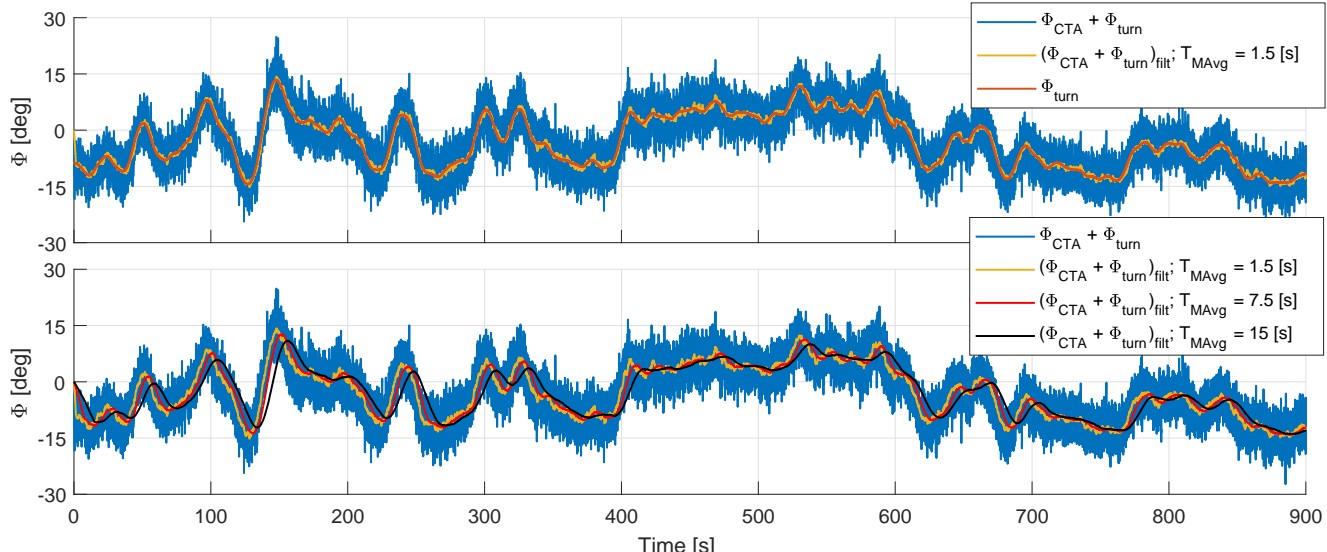

**Figure 6.** Top: time history of the combined wind direction changes experienced by the farm (blue), its 1.5 s moving average (orange), and the measured turntable rotation (red). Bottom: effect of the increasing time averaging window $T_{\mathrm{MAvg}}$ on the wind direction time series.

## 3.1 Wind farm models

Three different wind farm models of different fidelity were used for the synthesis of the LUTs: the lower level of fidelity is provided by the FLORIS model, described in §3.1.1, the intermediate level by a data-augmented version of FLORIS, described in §3.1.2, while the higher fidelity level is given by a purely data-driven model, described in §3.1.3.

For consistency with the wind tunnel experiments, a wind direction change was accounted for in the models as a rotation of the wind farm. A variation of the wind direction is therefore also associated with a slight variation of the ambient speed sensed by each wind turbine, because of the horizontal shear of the inflow shown in Fig. 3. The extra velocity component caused by the motion of the turbine and its effect on the local wind direction were not included in the models because they are negligible.

### 3.1.1 FLORIS model

Given a set of ambient wind conditions, the FLORIS model computes the steady-state flow within a wind farm and, in turn, the power output of the individual turbines (Doekemeijer et al., 2018). The present results were obtained with the implementation available online (Doekemeijer and Storm, 2018), using the *selfSimilar* velocity deficit, the *rans* deflection, the wake model of Bastankhah and Porté-Agel (2016), the *quadraticRotorVelocity* wake combination, and the *crespoHernandez* added turbulence (Crespo and Hernández, 1996). To improve accuracy at the cost of a slightly increased computational effort, the power of a

turbine is computed by integrating the flow at the rotor disk using $P = 1/2\rho \int_A V^3 C_{\mathrm{P}} \mathrm{d}A$ (where $\rho$ is air density, $V$ the local wind speed and $A$ the rotor disk area), instead of the original implementation based on the rotor-average wind speed. The speed-dependency of the thrust and power coefficients and the yaw-dependent power losses reported in Fig. 2 were implemented as





well. The ambient wind field in the model is horizontally sheared to match the wind tunnel inflow. The model was tuned based on wake measurements of one isolated G1 turbine, as discussed in Campagnolo et al. (2019), obtaining the parameters reported in Table 1. Accordingly, the wind speed at $y = 0$ was set to $5.25$ m/s, while the turbulence intensity was set to $6.1\%$.

**Table 1.** FLORIS parameters calibrated according to Campagnolo et al. (2019).

| $\alpha$ | $\beta$ | $k_{\mathrm{a}}$ | $k_{\mathrm{b}}$ | $TI_{\mathrm{a}}$ | $TI_{\mathrm{b}}$ | $TI_{\mathrm{c}}$ | $TI_{\mathrm{d}}$ |
|---|---|---|---|---|---|---|---|
| 0.9523 | 0.2617 | 0.0892 | 0.027 | 0.082 | 0.608 | $-0.551$ | $-0.2773$ |

### 3.1.2 Data-augmented FLORIS model

An improved level of fidelity is obtained by an augmented version of the baseline FLORIS model (termed FLORIS-Augm), following the approach described in Schreiber et al. (2019). The central idea of model augmentation is to surgically insert additional terms into the governing equations to represent expected errors or effects lacking in the model (for example, secondary steering, which is very relevant in the present context). The correction terms are expressed in terms of parametric functions that are identified (or learned) from operational data. Since a baseline performance is provided by the underlying FLORIS

model, learning is limited to small errors, which somewhat eases the requirements on the data. On the contrary, a purely data-driven approach, which does not use a reference model as a baseline, poses more stringent requirements on the training dataset; indeed, a data-driven model only "knows" what is in the data and nothing else. In practical field applications, it is possibly difficult to generate a rich-enough dataset to identify a model of high quality and wide generality.

The model augmentation method was demonstrated with the use of standard SCADA (Supervised Control and Data Ac-

quisition) data in Schreiber et al. (2019). Here, a similar approach was followed, by adding to FLORIS correction terms for non-uniform inflow and secondary steering (Fleming et al., 2018). The errors were then identified based on the power output of the three turbines in a variety of conditions, including different wind directions and different yaw misalignments, using a subset of the data used for the derivation of the Data-Driven model described in the following. Further details are given in Schreiber et al. (2019).

Although the FLORIS-Augm model is more accurate than the baseline FLORIS, it is still not perfect. Therefore it is interesting to verify if an even higher fidelity model might improve the performance of the wind farm controller. To answer this question, yet another unique ability of wind tunnel testing was exploited here. An extensive, high-quality dataset covering all operating conditions of interest was obtained in the wind tunnel. Based on this dataset, a high-fidelity, purely data-driven model is derived next.

### 25 3.1.3 Data-Driven model

A dataset was generated by measuring the power output of the three turbines for the 11 wind directions $\Phi = [0, \pm 2.29, \pm 4.58, \pm 6.89, \pm 9.21, \pm 11.54]$ deg. For each wind direction, the two upstream turbines were operated at various steady misalignment





angles $\gamma$ in the range $\pm 10$ deg around the optimal misalignments that, according to the FLORIS model, maximize the total plant power.

The data-driven model was obtained by best-fitting a response surface to the resulting set of data points, using shape functions inspired by experimental observations and the wake superimposition models used in FLORIS. The formulation of the interpolating shape functions is presented in appendix A.

### 3.1.4 Normalized power

The normalized power $P_{\mathrm{n},j}$ of the $j$-th wind turbine is defined as

$$P_{\mathrm{n},j} = \frac{P_j}{1/2\rho A U_j^3}, \tag{2}$$

where $U_j$ is the ambient wind speed at the location of that turbine. Here, the ambient speed is measured by the reference pitot tube, and then corrected for the tunnel horizontal shear. The total normalized wind farm power is defined as $P_{\mathrm{n,WF}} = \sum_j P_{\mathrm{n,j}}$.

For a turbine operating in undisturbed inflow, normalized power is equal to the standard power coefficient $C_{\mathrm{P}}$. However, normalized power and power coefficient differ for a turbine operating in the wake of an upstream machine. Normalized power is preferred to the power coefficient in the present analysis, because it reveals the reduced power extraction of a waked turbine when compared to an unwaked one, a difference that is lost to the classical power coefficient. In fact, two turbines —one in the wake of the other— might be operating in region II at the same power coefficient, although the downstream machine would have a much reduced power output than the front one.

### 3.1.5 Comparison of the three models

Figure 7 shows the normalized power of the individual turbines and the whole cluster for the case $\Phi = 0$ deg (i.e. with the wind blowing parallel to the row of turbines). Results are plotted versus the misalignment angles $\gamma_1$ and $\gamma_2$ of the two front turbines WT1 and WT2. Measured data points are indicated with red dots, while smooth surfaces show the predictions of the baseline FLORIS (left), FLORIS-Augm (center) and Data-Driven (right) models. A quantitative overall measure of the quality of the fits is given by the root mean square (RMS) errors $e_{\mathrm{RMS}}$, expressed in percent of the available freestream wind power and included in the legends.

By looking at the plots and at the fitting RMS errors, it appears that the quality of the models degrades when moving downstream along the row of turbines, as expected, considering the increasing role of wake interactions. A comparison of the plots by column reveals the increasing level of fidelity of the models, where FLORIS-Augm is better than FLORIS, and Data-Driven is better than FLORIS-Augm.

### 3.2 Look-up table computation

In general, the LUTs for an open-loop wake steering controller should depend on wind direction, wind speed and turbulence intensity (because of its effect on wake recovery). However, in the present wind tunnel experiments the last two parameters are



(a) WT1 normalized power.

(b) WT2 normalized power.

(c) WT3 normalized power.

(d) Wind farm normalized power.

**Figure 7.** Normalized power of the individual turbines and of the wind farm, as function of the misalignment angle $\gamma$ of the two front turbines WT1 and WT2, for the wind direction $\Phi = 0$ deg. Red dots: experimental measurements. Smooth surfaces: baseline FLORIS (left), FLORIS-Augm (center), and Data-Driven (right) models.



kept constant, so that the LUTs were scheduled only with respect to wind direction. A resolution of 0.2 deg was used for wind directions $\Phi \in \pm 2$ deg, whereas a lower resolution of 1 deg was used outside of this range.

For robustness, wind direction and yaw uncertainties should be taken into account in the calculation of the LUTs (Quick et al., 2017; Rott et al., 2018; Simley et al., 2019). Here only uncertainties in wind direction were considered, because yaw
uncertainties due to possible sensor errors are negligible for the calibrated G1s.

Steady-state models as the ones used in this work already include the effects of the higher frequency wind direction changes of the spectrum. For example, the wake profiles measured by Campagnolo et al. (2019) and used to identify the model parameters of Table 1 represent mean steady values, while the actual instantaneous wake undergoes meandering fluctuations. In this sense, it is important to realize that the wake model already contains the effects of the wind direction changes naturally present
in the wind tunnel flow, whose spectrum is reported in the top plot of Fig. 4 in red. However, steady models lack the flow dynamics at the lower frequencies and the delays caused by the advection downstream with a finite travel speed. These models are therefore only capable of predicting slow changes of wind turbine power (Simley et al., 2019). A robust control formulation (Rott et al., 2018) should take into account the uncertain knowledge of the wind direction at these slower time scales.

Here again, wind tunnel testing presents some opportunities that are hardly available when testing in the field. In fact, the
actual turntable rotation represents the "ground truth", while the controller takes as input the filtered signal (shown in the lower part of Fig. 6). It follows that wind direction uncertainties are known in this case and are represented by the difference $\Delta\Phi$ between these two quantities. Therefore, one can change the value of the uncertainties (which is challenging in reality at full scale, since the ground truth is typically unknown) by simply changing the filtering of the turntable rotation. This approach was used here to study the effects that uncertainties have on the performance of the controller. Figure 8 reports the distribution
of $\Delta\Phi$ for two values of $T_{\mathrm{MAvg}}$ equal to 7.5 and 15 s. The fitted Gaussian normal distributions have standard deviations $\sigma_\Phi = 2.01$ deg and 3.42 deg, respectively. For $T_{\mathrm{MAvg}} = 1.5$ s wind direction uncertainties are negligible.

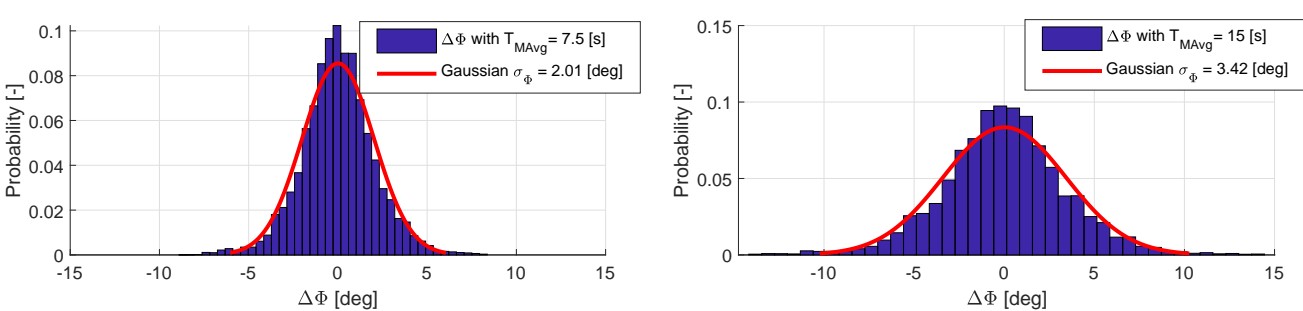

**Figure 8.** Distribution of wind direction uncertainties, i.e. difference between the actual turntable position and its filtered value (which is the wind direction input to the controller) for $T_{\mathrm{MAvg}} = 7.5$ s (left) and $T_{\mathrm{MAvg}} = 15$ s (right).

For each flow model, robust LUTs were computed based on the approach of Rott et al. (2018) for $\sigma_\Phi = [0 : 2 : 6]$ deg. The Matlab pattern-search algorithm was used to solve the resulting bounded optimization problem. For each considered wind



direction $\Phi$, the optimal yaw misalignments $\gamma_1^*$ and $\gamma_2^*$ for WT1 and WT2 were computed as

$$[\gamma_1^*(\Phi), \gamma_2^*(\Phi)] = \arg\max_{\gamma_1, \gamma_2} \sum_{k=0}^{9} P_{\mathrm{M}}\left(\Phi + \Delta\Phi_k, \gamma_1 - \Delta\Phi_k, \gamma_2 - \Delta\Phi_k, -\Delta\Phi_k\right) f(\Delta\Phi_k), \tag{3a}$$

$$\text{such that: } [\gamma_1, \gamma_2] \in \pm 30 \text{ deg}, \tag{3b}$$

where $\Delta\Phi_k = (4k/9 - 2)\sigma_\Phi$ is the wind direction uncertainty varying in the range $\pm 2\sigma_\Phi$, $f \sim \mathcal{N}(0, \sigma_\Phi)$ is the Gaussian

normal distribution and $P_{\mathrm{M}}(\Phi, \gamma_1, \gamma_2, \gamma_3)$ the wind farm power predicted by the wind farm model.

The LUTs obtained with the baseline FLORIS model for different values of $\sigma_\Phi$ are shown in Fig. 9a. The effect of an increasing uncertainty is that of generating a smoother transition around $\Phi = 0$ deg, and in general smaller misalignment of the turbines with respect to the wind.

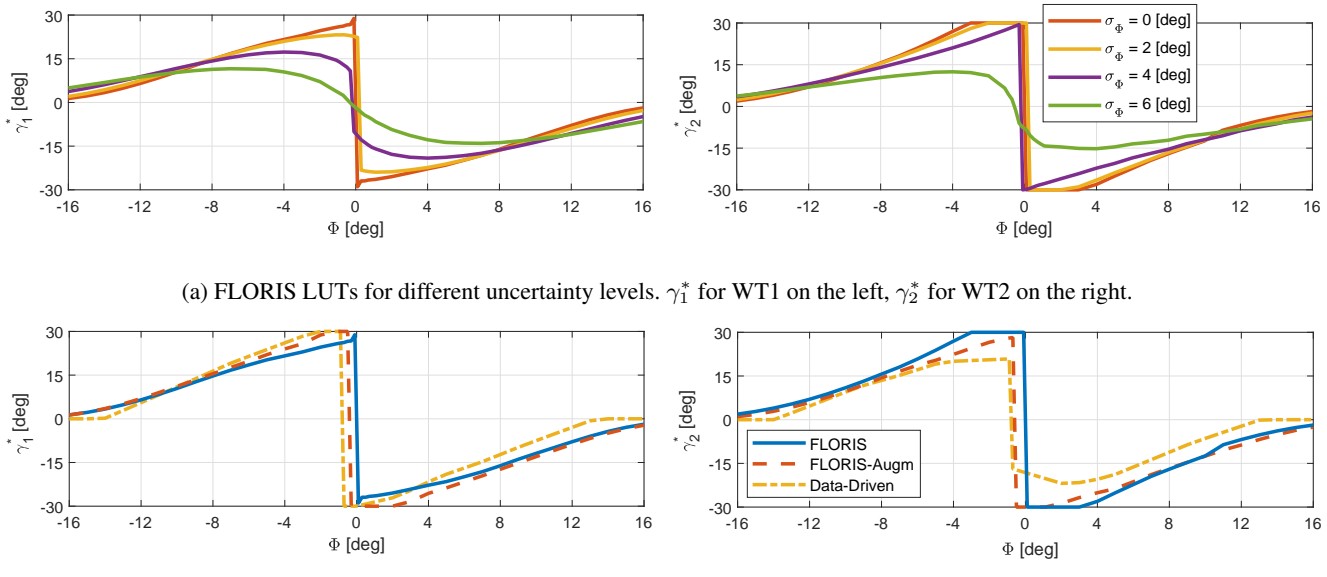

(a) FLORIS LUTs for different uncertainty levels. $\gamma_1^*$ for WT1 on the left, $\gamma_2^*$ for WT2 on the right.

(b) LUTs for different models and $\sigma_\Phi = 0$ deg. $\gamma_1^*$ for WT1 on the left, $\gamma_2^*$ for WT2 on the right.

**Figure 9.** LUTs of optimal misalignment angles $\gamma_1$ and $\gamma_2$ vs. wind direction $\Phi$.

Figure 9b compares the LUTs obtained with the three models for $\sigma_\Phi = 0$ deg.

Considering the front turbine misalignment $\gamma_1$ (left plot of Fig. 9b), the main difference among the LUTs is in the position of the transition point between positive and negative yaw offset, which is 0, $-0.5$ and $-0.8$ deg for the baseline FLORIS, FLORIS-Augm and Data-Driven models, respectively. The non-zero transition point predicted by two of the models can be ascribed to the non-symmetric behavior of power for the cluster of turbines, shown in the left plot of Fig. 10 for the greedy policy, i.e. no wake steering control. Indeed, the figure shows that the minimum of the wind farm normalized power is at

about $-0.8$ deg, i.e. for a wind blowing slightly from the right of the row of turbines when looking upstream. This is due to



the combined effects of the tunnel horizontal shear and the slight lateral deflection for null yaw misalignment created by the vertically sheared flow.

Looking at the second turbine misalignment $\gamma_2$ (right plot of Fig. 9b), there is a significant difference among the three models. In fact, the baseline FLORIS does not include secondary steering, which is on the other hand represented to a different
level of fidelity by the FLORIS-Augm and Data-Driven models. This effect leads to smaller misalignments for the second compared to the front machine, in agreement with other recent wind tunnel studies (Campagnolo et al., 2016c; Bastankhah and Porté-Agel, 2019).

## 4 Results

### 4.1 Maximum theoretical performance of the controllers

Before considering the behavior of the controllers in the experiments, it is interesting to establish a theoretical upper limit to their performance, neglecting dynamic effects, limited yaw rates and uncertainties. To this end, the Data-Driven model was used as plant, being essentially an exact representation (except for measurement errors) of the wind farm behavior for a constant mean wind speed. The wind farm power output was computed using the greedy control policy and the LUTs for $\sigma_\Phi = 0$ deg. The total power output of the cluster is shown in the left plot of Fig. 10, while the right plot shows the percent power gain with
respect to the greedy policy.

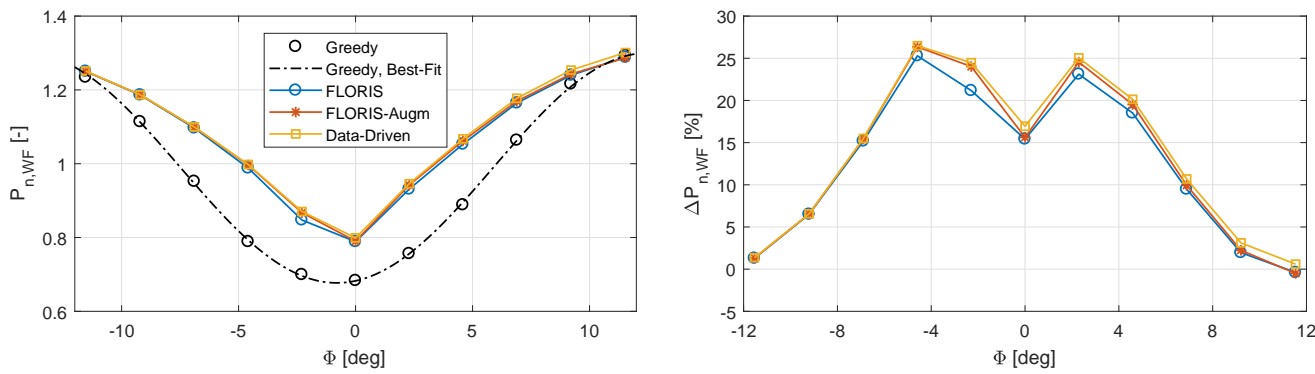

**Figure 10.** Left: wind farm power output as a function of wind direction for different control policies. Right: maximum theoretical percent power gain with respect to the greedy policy for the three flow models.

Results indicate that all models lead to positive gains for all investigated wind directions, up to about 25% in the best conditions. The gains for the baseline FLORIS model are only slightly smaller than for the FLORIS-Augm and the Data-Driven model. This appears to indicate that the cost function of problem (3) is rather insensitive to the details of the underlying model in the absence of uncertainties. However, these results might be misleading, because uncertainties are indeed present in
reality and play a significant role, as shown later.





The results of Fig. 10 can be used to compute the maximum possible performance of the controllers for the wind direction time series used in the experiments, and shown in Fig. 4. Under the assumption of an exact knowledge of the wind direction, an instantaneous realization of the required yaw misalignments, and in the absence of any flow dynamics, the power gains of the baseline FLORIS, FLORIS-Augm and Data-Driven LUTs are respectively equal to 10.73%, 11.41% and 11.84%. These

figures establish a non-achievable maximum theoretical performance of the controllers for this particular farm layout and wind direction time history.

## 4.2   Impact of different non-robust controller implementations

Next, wind tunnel tests were performed to characterize the effects of the following aspects of open-loop wake steering:

- Uncertainty level (which, in the present context, is related to the filtering of the wind direction, i.e. of the turntable
rotation);

- Effect of a finite yaw rate and of neglected wake dynamics;

- Model fidelity, according to the three considered models FLORIS, FLORIS-Augm and Data-Driven.

The analysis is conducted first for a non-robust controller implementation, i.e. for the formulation expressed by problem (3) with $\sigma_\Phi = 0$ deg, while the performance of a robust controller is considered later in the paper.

Dynamic changes in wind direction were obtained by actuating the wind tunnel turntable, as described in §2.3, in the offshore inflow conditions described in §2.2. Each test was performed for a total of 90 min divided into 9 intervals of 10 min each. This allowed for the periodic calibration of the wind tunnel and the wind turbine sensors, to guarantee the highest possible accuracy of the measurements.

Tests with the greedy control strategy were repeated four times, dispersed over the course of the experimental campaign.

The averaged power values for the 90 min wind direction time series were normalized with the results of the first test, and are shown in Fig. 11 for the whole wind farm and for each wind turbine. The standard deviation of these values across the four repetitions is equal to 0.98% of the available freestream wind power for the whole cluster, and to 0.15%, 0.42% and 0.51% for WT1, WT2 and WT3, respectively. These uncertainties, which can be mainly ascribed to errors of the pitot transducer and shaft torque-meter, are acceptable considering the purpose of this analysis and are well below the differences caused by the

various effects studied herein.

### 4.2.1   Effect of wind direction uncertainties

Experimental tests were performed with non-robust LUTs obtained from the baseline FLORIS model for the three filtering values $T_{\mathrm{MAvg}} = 1.5$ s ($\sigma_\Phi = 0$ deg), 7.5 s ($\sigma_\Phi = 2.01$ deg), and 15 s ($\sigma_\Phi = 3.42$ deg), which correspond to the three wind direction time histories shown in the lower plot of Fig. 6.

The power gains with respect to the greedy policy are shown in Fig. 12. Average values aggregated over the whole wind direction time history are shown at wind farm level and for the single turbines. As expected, results indicate a progressive



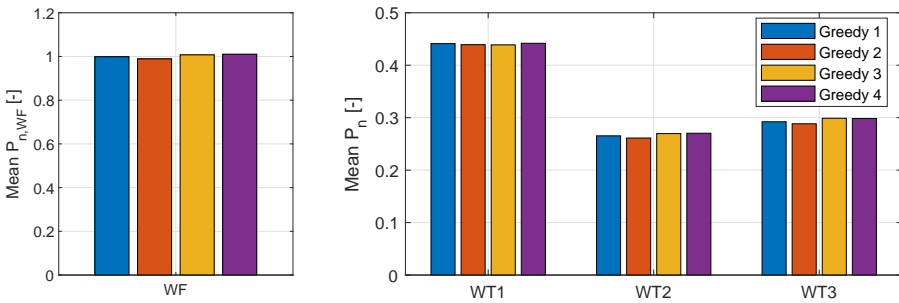

**Figure 11.** Repeatability of the experiments: averaged, normalized power for the wind farm (left) and the individual wind turbines (right) for four 90 min repetitions with the greedy control strategy.

degradation of performance for an increasing level of uncertainty (i.e. for increasing $T_{\mathrm{MAvg}}$ and hence $\sigma_\Phi$). The effects on the front turbine are very limited, whereas they are more pronounced for the second and the third turbine due to the effects caused by wake interactions. Indeed, power variations at the front turbine caused by a non-exact alignment with the wind are rather small according to the cosine law shown in the lower part of Fig. 2; on the other hand, a non-exact misalignment has a
5   much amplified effect on the location of the wake downstream of the rotor, which may induce large losses on the downstream turbines. Such losses could be even larger for a greater spacing between turbines than the 5D of the present experiment. The impact on the overall farm power output is substantial: increasing $\sigma_\Phi$ from 0 to 3.42 deg cuts the power gain by more than half.

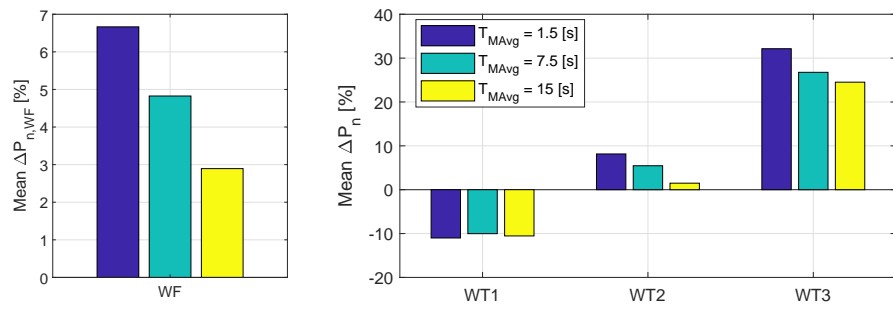

**Figure 12.** Averaged power gains aggregated over all wind directions for different values of $T_{\mathrm{MAvg}}$ (i.e. of wind direction uncertainties $\sigma_\Phi$).

### 4.2.2 Effect of yaw rate and neglected wake dynamics

Even in the absence of wind direction uncertainties ($T_{\mathrm{MAvg}} = 1.5$ s, $\sigma_\Phi = 0$ deg), the farm-level power gain (about 6.7%,
10   Fig. 12) is much lower than the established theoretical upper limit (10.73%, §4.1). This difference is caused by the limited yaw rate of the turbines and by having used a steady model and controller, which implies neglecting the dynamics of wake





interaction (including the intrinsic dynamics of the wake, its slow-scale meandering fluctuations, and the advection downstream of any change with a finite travel speed).

Figure 13 establishes the impact of these effects on the performance of the controller. The plot reports plant-level power gains with respect to the greedy policy, as functions of wind direction. To reduce noise in the figure, each point in the plot represents

the average power gain for a wind direction bin with a width of 2.5 deg. The solid orange line with $*$ symbols reports the gains measured in the experiment. The solid blue line with $\circ$ symbols indicates the theoretical upper limit when using the baseline FLORIS model, obtained by binning the data shown in the right plot of Fig. 10. The dash-dotted green line with $\triangle$ symbols shows the gains computed by a simulation conducted with the Data-Driven model, using the yaw misalignment angles $\gamma^{\text{Meas}}$ measured in the experiment. Since the Data-Driven model can be assumed to be an exact steady-state representation of the

experiment, the green line of the figure shows the impact of a limited yaw rate on the maximum, theoretical performance. Finally, the dashed red line with $\square$ symbols shows the gains when using the yaw misalignment angles $\gamma^*$ requested by the controller, computed with the Data-Driven model, i.e. without considering limits in the yaw rate.

These curves allow for the quantification of the following effects:

– The difference between the lower orange curve and the green curve can be attributed to neglected wake dynamics; this

non-negligible difference could in principle —at least in part— be regained by using a dynamic controller, instead of the steady-state controller considered here.

– The difference between the green and the red curves is due to a finite yaw actuation rate. This difference indicates that another non-negligible power capture improvement could be gained by a faster actuation, which however would have to be traded against increased loading and actuator duty cycle (ADC). This gain is limited to relatively small misalignments

(about $\Phi \in \pm 6$ deg in the figure).

– Finally, the difference between the red and the upper blue curve is due to remaining effects, such as the finite sampling time of the controller. This small difference indicates that these effects are negligible with respect to the others.

### 4.2.3 Effects of wind farm model fidelity

The influence of wind farm models with an increasing level of fidelity was assessed for the case of negligible wind direction

uncertainties ($T_{\text{MAvg}} = 1.5$ s, $\sigma_\Phi = 0$ deg). Figure 14 shows the averaged wind farm power gains aggregated over the whole time history for the three different models. The experimentally measured gains are reported in Fig. 14a, while Fig. 14b shows the gains obtained by simulations with the Data-Driven model as plant and the misalignment angles $\gamma^{\text{Meas}}$ measured in the experiments. The maximum theoretical power gains of §4.1 are shown in Fig. 14c. Again, the lower gains of Fig. 14b compared to Fig. 14c can be attributed to the limited yaw rate. The lower power gains of Fig. 14a compared to Fig. 14b can be attributed

to neglected dynamics.

The figures show that LUTs synthesized with better wind farm models lead to higher power gains. In fact, for the wind tunnel experiments, employing the FLORIS-Augm and Data-Driven LUTs increases the power gain by 5.1% and 16.7%, respectively,





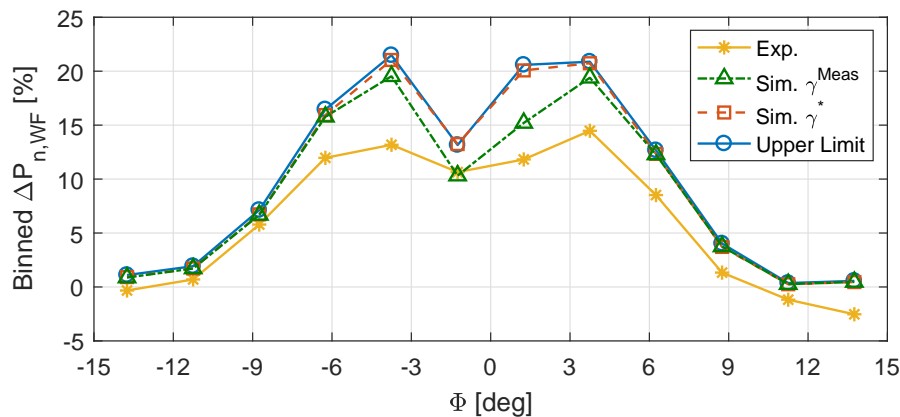

**Figure 13.** Wind farm power gains with respect to the greedy policy vs. wind direction $\Phi$. Tests were performed with zero wind direction uncertainty ($\sigma_\Phi = 0$ deg) and LUTs synthesized from the baseline FLORIS model.

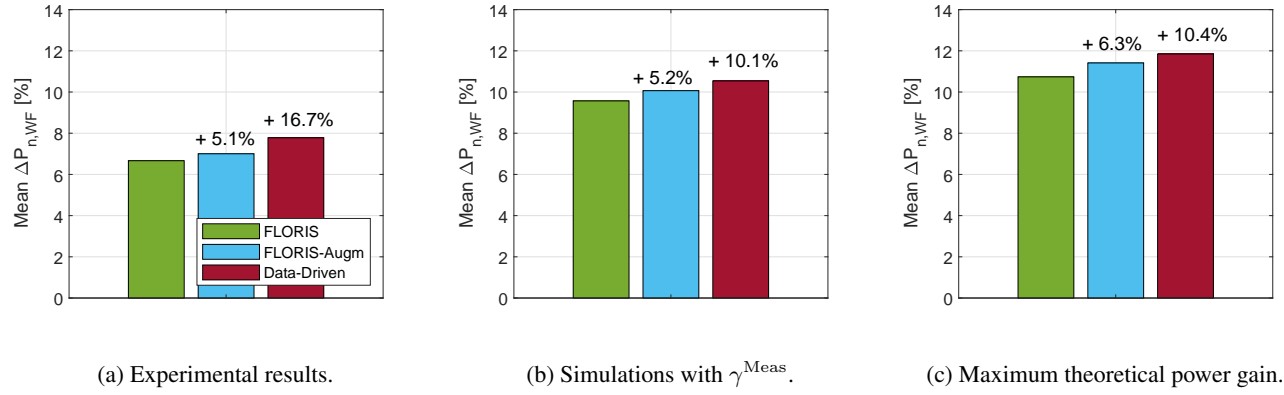

(a) Experimental results.     (b) Simulations with $\gamma^{\mathrm{Meas}}$.     (c) Maximum theoretical power gain.

**Figure 14.** Normalized wind farm power gains for the three models of different fidelity.

compared to the baseline FLORIS case. The simulation results of Figs. 14b and 14c show a similar trend. However, the benefits of the highest fidelity model over the lower fidelity ones for both simulation cases (10.1% and 10.4%) are smaller than in the experiments (16.7%). This might be due to dynamic effects, which could affect the controller performance in different ways depending on the underlying flow model.

5   **4.3   Robust implementation accounting for wind direction uncertainties**

Further experiments were conducted using robust LUTs computed according to problem (3) for $\sigma_\Phi = [0 : 2 : 6]$ deg based on all three models. For all tests, the wind direction (i.e. the turntable encoder signal) was filtered with a moving average with $T_{\mathrm{MAvg}} = 7.5$ s. This corresponds to 10 min at full scale, similarly to typical 10 min SCADA data. As shown in Fig. 8, this means that the simulated wind direction uncertainty in the experiments had a standard deviation $\sigma_\Phi = 2.01$ deg.



Figure 15 reports the power gains with respect to the greedy case for the baseline FLORIS model for varying uncertainty levels $\sigma_\Phi$ in the formulation of the LUTs (i.e. for increasing robustness). To reduce noise, the plot was generated with average values according to wind direction bins with a width of 2.5 deg. The figure shows that, with an increasing level of uncertainty, power is shifted from the most downstream machine (bottom left plot) to the upstream one (top left plot), whereas the turbine

in between is substantially unaffected (top right plot). This makes intuitive sense: with large uncertainties in the wind direction, the power output of downstream machines becomes more uncertain; therefore, the controller tries to lose less power upstream, where changes in wind direction have a more limited impact on the local capture. This clearly comes at a cost, and the total power output at farm level decreases (bottom right plot). With small uncertainties, the opposite happens: since the location of the wakes is more certain, it pays off to deflect the wake of the front machines in order to try to boost capture downstream.

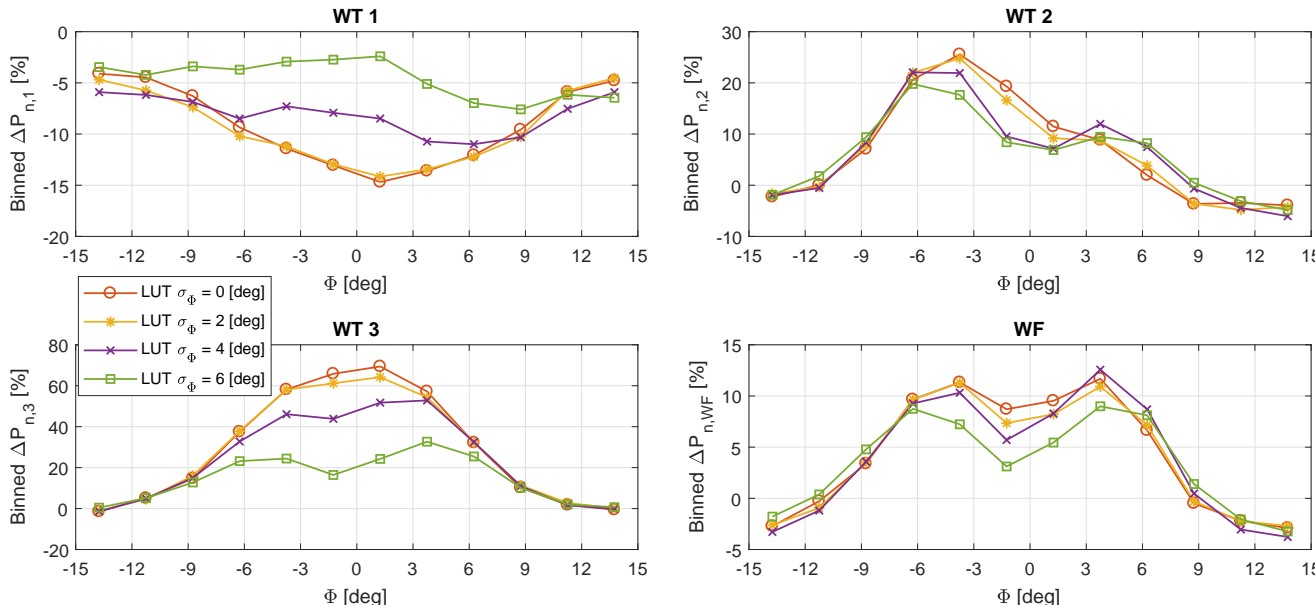

**Figure 15.** Power gains with respect to the greedy policy vs. wind direction, shown for the baseline FLORIS model and increasing robustness of the LUTs.

It should also be noted that wind farm power gains may be negative away from conditions with strong wake interactions. This is indeed the case here for wind directions $\Phi < -10$ deg and $\Phi > 8$ deg. This suggests that wake steering should only be applied in cases where strong enough interactions are expected, and switched off elsewhere.

Figure 16a shows the overall experimental power gains with respect to the greedy case for the various models and for increasing robustness. Additionally, Fig. 16b and Fig. 16c report simulation results with the Data-Driven model as plant and

the effectively realized misalignment angles $\gamma^{\text{Meas}}$ or the demanded misalignment angles $\gamma^*$, respectively. The power gain change with respect to the FLORIS LUTs with $\sigma_\Phi = 0$ deg is reported above each column. If one looks at the experimental data, shown in Fig. 16a, the power gains are equal to about $4-6\%$, a range that is considerably lower than the theoretical





maximum reported in Fig. 14c. Moreover, gains are higher and less affected by uncertainties for the better fidelity model. From this point of view, it appears that a higher fidelity model could provide better and more robust results than a lower fidelity one.



(a) Experimental results.

(b) Simulations with $\gamma^{\mathrm{Meas}}$.

(c) Simulations with $\gamma^*$.

**Figure 16.** Overall power gains for the three models of different fidelity and increasing robustness of the LUTs.

The situation considered here is indeed much more realistic than the one discussed in §4.2, and the lower gains observed in this case are due to wind direction uncertainties, limited turbine yaw rate and wake propagation dynamics. The best perfor-
5   mance in the experiments is obtained with the Data-Driven model for $\sigma_\Phi = 6$ deg. Additionally, the experimental results of the FLORIS-Augm model are better than the ones of the baseline FLORIS. For both the Data-Driven and the FLORIS-Augm models, lower gains are obtained when neglecting uncertainties ($\sigma_\Phi = 0$ deg), which points to the importance of using a robust formulation. Surprisingly, the baseline FLORIS model exhibits just the opposite behavior.





The maximum gains in the experiments are obtained for $\sigma_\Phi = 4$ and 6 deg for the FLORIS-Augm and Data-Driven models, respectively. These values are significantly higher than the actual uncertainty in the wind direction signal (equal to $\sigma_\Phi = 2.01$ deg). This is probably due to the limited yaw rate. In fact, Fig. 16b shows that with a limited rate even the simulation results yield the best gains for $\sigma_\Phi = 4$ deg, while Fig. 16c shows that without rate limits the optimal performance is obtained

for the effective uncertainty $\sigma_\Phi = 2$ deg present in the driving signal. This makes intuitive sense: LUTs computed with a lower uncertainty result in higher gradients of the misalignment angle with respect to wind direction changes, which are less likely to be achieved by a limited yaw rate.

These results allow for some interesting considerations. First, if the model is strongly biased, as in the present baseline FLORIS case, introducing robustness in the formulation may decrease performance. This is in contrast to the results reported

by Rott et al. (2018), who, however, did not consider biased models. On the contrary, robustness increases performance if the underlying models have better fidelity, which is the case here for the FLORIS-Augm and Data-Driven models. Moreover, the impact of a limited yaw rate should certainly be taken into account in the calculation of the LUTs, as proposed by Simley et al. (2019). More importantly, better models and robust LUTs lead to better performance.

### 4.4   Impact on actuator duty cycle and loads

The wind tunnel experiments were also used to evaluate the impact of wake steering on yaw control effort and fatigue loads.

The average wind farm yaw ADC is defined as

$$\mathrm{ADC_{WF}} = \frac{1}{N_{\mathrm{WF}}} \sum_{j=1}^{N_{\mathrm{WF}}} \frac{1}{T} \int\limits_0^T \frac{|\dot{\gamma}_{\mathrm{nac},j}(t)|}{\dot{\gamma}_{\mathrm{max}}}\, \mathrm{d}t, \tag{4}$$

where $\dot{\gamma}_{\mathrm{nac},j}(t)$ is the time rate of change of the orientation of the $j$-th wind turbine, and $N_{\mathrm{WF}} = 3$ the number of turbines. The average wind farm ADC is an indicator of the usage of the yaw actuators, and could therefore be used to quantify the impact

of wake steering control on the maintenance cost of the yaw drives.

Figure 17 shows the increase $\Delta\mathrm{ADC_{WF}}$ with respect to the greedy control policy. The effect of filtering the wind direction signal is shown in Fig. 17a for the non-robust LUT baseline FLORIS formulation. As expected, a longer averaging window smooths the signal, resulting in less yaw activity (but also less power and more fatigue damage, as shown in Fig. 12 and 18a). The increase in ADC with respect to the greedy control case is, however, very substantial.

The results obtained with robust LUTs based on the three models are shown in Fig. 17b, for a wind direction signal filtered with $T_{\mathrm{MAvg}} = 7.5$ s. Comparing Fig. 17b with Fig. 17a shows that a robust formulation decreases ADC, as expected by the reduced misalignments prescribed by the controller (cf. Fig. 9a). Increasing robustness has a dramatic effect on ADC, which however is still much higher than in the greedy case even for $\sigma_\Phi = 6$ deg. There is a clear tradeoff in wake steering between the benefits of an improved power capture and the detriments caused by an increased ADC. Additionally, the figure also shows

that model fidelity has only a relatively minor effect on ADC.

Damage Equivalent Loads (DELs) were computed from bending moments measured on the rotating shaft and at the tower base. Load signals were first filtered above the 6P rotor frequency to remove high-frequency mechanical vibrations. In addition,





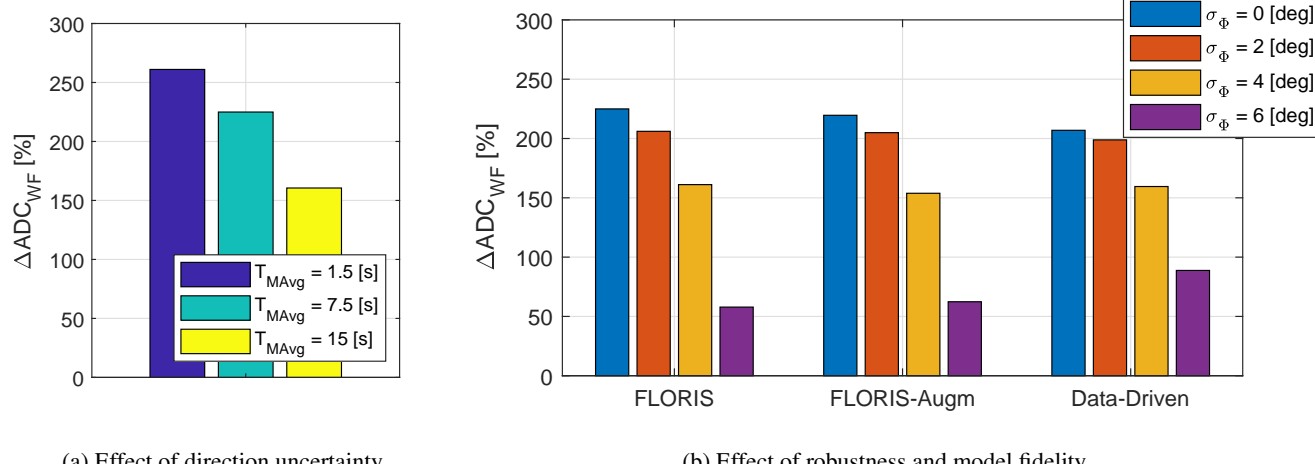

(a) Effect of direction uncertainty.

(b) Effect of robustness and model fidelity.

**Figure 17.** Change in wind farm yaw ADC with respect to the greedy case. Left: non-robust LUT for the baseline FLORIS model and varying $T_{\mathrm{MAvg}}$. Right: LUTs for the three models and increasing robustness, for a wind direction signal filtered with $T_{\mathrm{MAvg}} = 7.5$ s.

tower loads were corrected from 1P harmonics generated by the small inertial and aerodynamic imbalance of each rotor. A similar correction was applied to the fixed-frame hub loads computed from the rotating shaft components. Once cleaned of the 1P component, the fixed-frame loads were projected back onto the shaft frame, obtaining rotating loads corrected for rotor imbalance.

Bending DELs of the rotating shaft are reported in Fig. 18, while tower base bending DELs are given in Fig. 19. In both cases, combined DELs were obtained by projecting the two measured orthogonal bending components on the direction associated with the maximum DEL, and normalizing by the temporal average of $1/2\rho\pi R^3 U_{\mathrm{Pitot}}^2$, where $R$ is the rotor radius. The loads for WT1 for varying $T_{\mathrm{MAvg}}$ and for the baseline FLORIS cases are not reported in the figure, due to a problem with the recording of the rotor azimuth of that turbine during these tests.

A few observations can be made looking at the results for the shaft DELs. First, as expected and as clearly visible in Fig. 18a, load mitigation with non-robust LUTs worsens rapidly for increasing uncertainty (i.e. increasing $T_{\mathrm{MAvg}}$), since more wake interactions are taking place downstream. Second, when using robust LUTs, model fidelity seems to have only a modest effect on DELs, as shown in Fig. 18b. Third, by pointing the rotor away from the wind, the DELs of the front machine have a moderate increase, which is again an expected behavior. However, it is particularly interesting to look at the effect of varying

robustness. Indeed, only a marginal increase of DELs is observed for $\sigma_\Phi = 6$ deg, which still corresponds to significant power gains (cf. Fig. 16). Moreover, wake steering is particularly beneficial for the DELs of the second and third turbines, with reductions varying between $7 - 12\%$, depending on the LUTs. In general, DEL reductions seem to be correlated with power gains: robust LUTs with the largest power gains also generate the maximal load reductions.

    Similar conclusions can be drawn from looking at the results for the tower base DELs, despite some differences compared

to the shaft loads. Although absolute loads on the front turbine never exceed those of the downstream ones (Campagnolo et al.,





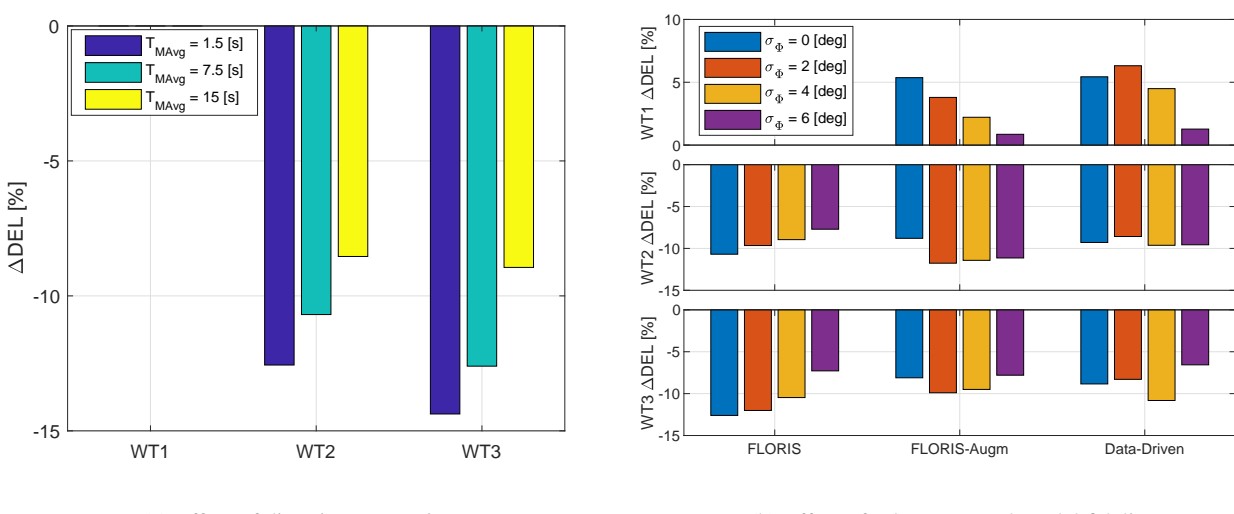

(a) Effect of direction uncertainty.

(b) Effect of robustness and model fidelity.

**Figure 18.** Change in combined rotating shaft DELs with respect to the greedy case. Left: non-robust LUT for the baseline FLORIS model and varying $T_{\text{MAvg}}$. Right: LUTs for the three models and increasing robustness, for a wind direction signal filtered with $T_{\text{MAvg}} = 7.5$ s.

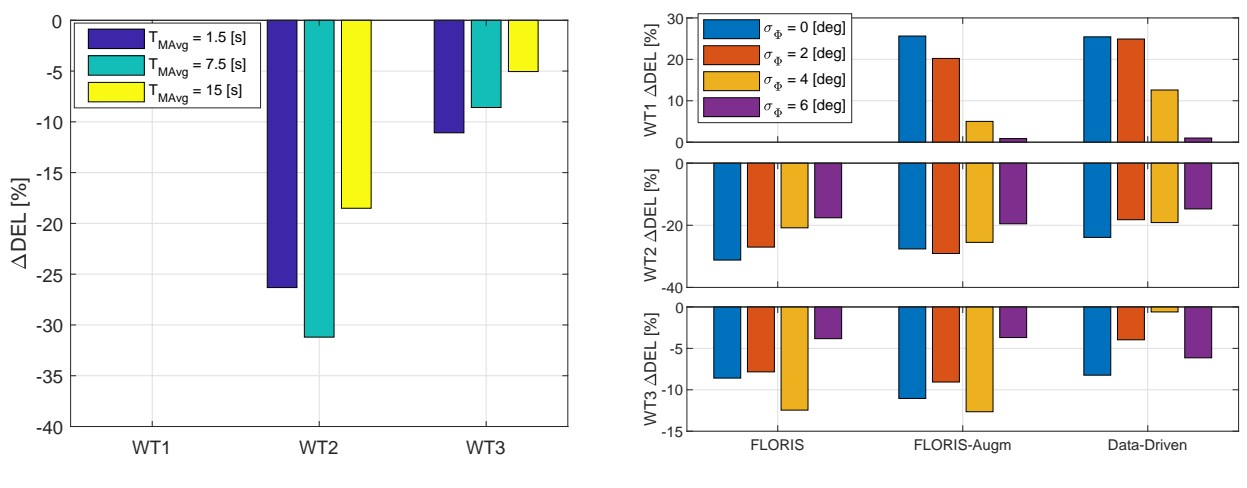

(a) Effect of direction uncertainty.

(b) Effect of robustness and model fidelity.

**Figure 19.** Change in combined tower base DELs with respect to the greedy case. Left: non-robust LUT for the baseline FLORIS model and varying $T_{\text{MAvg}}$. Right: LUTs for the three models and increasing robustness, for a wind direction signal filtered with $T_{\text{MAvg}} = 7.5$ s.




2020), the tower DELs of WT1 increase much more significantly with yaw misalignment than the shaft DELs (compare top plot of Fig. 19b with the one of Fig. 18b). Again the increment becomes almost negligible when robust LUTs computed with $\sigma_\Phi = 6$ deg are used, as shown in Fig. 19b. The tower DELs of the second turbine are significantly reduced, up to about 30% depending again on the LUTs and on the filtering of the wind direction, whereas the load mitigation on the third turbine is less

pronounced and shows a less clear trend.

## 5   Conclusions

This paper has presented an analysis of the effects of wind direction changes on the performance of an open-loop wake steering controller.

The study was based on the results of a new unique set of experiments conducted with three scaled turbines operated in

a large boundary layer wind tunnel. Wind direction changes were simulated with a turntable, driven by actual measurements performed in the field that were scaled to match the accelerated time of the experiment. The filtered wind direction provided as input to the controller was shown to represent a realistic approximation of the signal that could be acquired by a met-mast in the field. Three different models of increasing fidelity were used for the synthesis of the control laws. The control formulation was based on an established robust approach, which includes a naive deterministic optimization as a special case.

The unique possibilities offered by testing in the known, repeatable and controllable environment of the wind tunnel were exploited here to:

– Establish a theoretical upper limit to the performance of the controller in the absence of dynamics;

– Separate the effects of neglected dynamics, model fidelity and actuation rate;

– Feed to the controller a variable level of uncertainty, in order to quantify its effects on performance.

Based on the results of this study, the following conclusions can be drawn:

– Higher fidelity models lead in general to slightly better results in terms of power capture, whereas the effects of fidelity on actuator usage and fatigue loads are modest. In addition, higher fidelity models appear to be less susceptible to the effects of uncertainties.

– The use of a robust formulation is beneficial in terms of power capture, but yields even higher payoffs when looking
25        at other metrics. In particular, the overall plant-level ADC and the DELs of the front turbine are greatly reduced when compared to a non-robust formulation.

– The previous statement is however only true if the underlying flow model is accurate enough. In fact, the use of a robust formulation actually decreased performance for the baseline FLORIS model (which lacks important effects such as secondary steering), both in terms of power capture and load mitigation downstream. This seems to indicate that
30        excessively simplified models should probably be avoided.



– Increasing the robustness of the controller has the effect of shifting power upstream, as the position of the wakes is affected by larger uncertainties than the ones caused on the front turbine by a non-exact alignment with the wind. This however comes at a cost, as higher wake interactions are allowed to take place for increasing robustness, in turn leading to a lower power capture at the plant level.

5 – A robust implementation may lead to power losses in conditions with weak or absent wake interactions. This might suggest the use of wake steering only around conditions where significant wake effects are expected, whereas it should be switched off elsewhere.

– There is a non-negligible margin in power capture performance that may be attributed to dynamic effects. This seems to indicate that dynamic controllers, as opposed to the steady-state ones used here, might lead to a better performance, at 10 the cost of a higher complexity.

– Yaw rate is an important performance driver, and indeed higher rates achieve better results in terms of power output at the farm level. However, this clearly comes at a large cost in terms of actuator usage and loading. Such tradeoffs can only be quantified by a system-level design study, which is however turbine and plant-dependent and beyond the scope of this paper.

15 The present work could benefit from improvements to the experimental setup and the control methods. A relatively straightforward modification to the turntable could allow for higher accelerations, filling a missing band of frequencies in the wind direction spectrum. Instead of using the turntable rotation as an approximation of a met-mast measured wind direction, the ambient conditions could be estimated directly from the wind turbine operational data (Schreiber et al., 2018). Finally, dynamic closed-loop controllers could be tested, to understand and quantify their potential benefits with respect to the present simpler 20 approaches.

## Appendix A: Interpolating functions for the Data-Driven surrogate model

A surrogate model of the behavior of the cluster of three turbines is derived based on experimental measurements of power and wake displacement, and on wake superposition principles.

Figure 20a reports the normalized power $P_{\mathrm{n},2}$ (cf. §3.1.4) of the second turbine in the row as a function of the wind direction 25 $\Phi$, when both turbines are aligned with the wind (i.e. for $\gamma_1 = \gamma_2 = 0$). The measured data points can be interpolated with the following function

$$\frac{P_{\mathrm{n},2}}{C_{\mathrm{P}}^{II}} = \begin{cases} 1 - B\left(1 - \sin\left(\pi C\Phi - D\right)\right), & \text{if } \frac{D-3/2\pi}{\pi C} < \Phi < \frac{D+\pi/2}{\pi C}, \\ 1, & \text{otherwise}, \end{cases} \tag{A1}$$

where $C_{\mathrm{P}}^{II}$ is the power coefficient below rated speed, while $B$, $C$, and $D$ are tunable parameters.





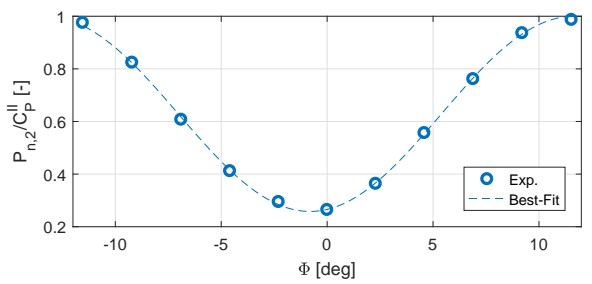
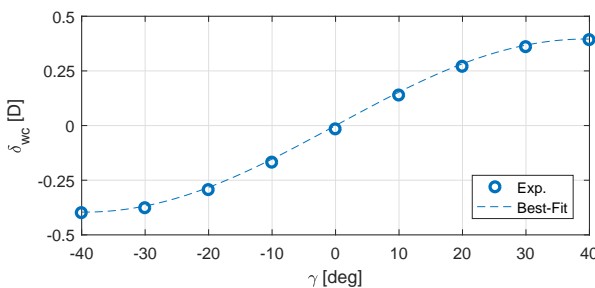

(a) Normalized power of WT2 vs. $\Phi$ for $\gamma_1 = \gamma_2 = 0$.      (b) Wake displacement vs. misalignment $\gamma$.

**Figure 20.** Experimental data points and their best fits for the derivation of the interpolating functions.

Figure 20b reports the lateral displacement $\delta_{\mathrm{wc}}$ of the wake of a G1 turbine as a function of the wind misalignment angle $\gamma$, measured 5D downstream of the rotor. The measured data points can be interpolated with the following function

$$\delta_{\mathrm{wc}} = E \sin\left(F\gamma\right), \tag{A2}$$

where $E$ and $F$ are tunable parameters. When the wake of an upstream turbine is deflected, the wake overlap at the downstream
machine can be approximated with the overlap that would occur for a wind direction $\Phi + \Delta\Phi$, where

$$\Delta\Phi \approx \sin\Delta\Phi = \frac{\delta_{\mathrm{wc}}}{\Delta X} = \frac{E\sin\left(F\gamma\right)}{\Delta X}, \tag{A3}$$

and $\Delta X$ is the longitudinal distance between the two turbines.

In region II, the power coefficient of a wind misaligned turbine can be expressed as

$$C_{\mathrm{P}} = C_{\mathrm{P}}^{II} \cos^n\left(\gamma + \phi\right), \tag{A4}$$

where $n$ is the power loss exponent, and $\phi$ is the phase asymmetry caused by a vertically sheared inflow.

These interpolating functions can be used to express the normalized power $P_{\mathrm{n},d}$ at a downstream machine as a function of the yaw misalignment of an upstream turbine $\gamma_u$ for a given wind direction $\Phi_0$. In fact, inserting Eq. (A3) into Eq. (A1), considering Eq. (A4), one gets

$$P_{\mathrm{n},d} = \begin{cases} C_{\mathrm{P}}^{II}\left(1 - B\left(1 - \sin\left(\pi\hat{C}\sin\left(F\gamma_u\right) - \hat{D}\right)\right)\right)\cos^{n_d}\left(\gamma_d + \phi_d\right), & \text{if } \frac{\hat{D} - 3/2\pi}{\pi\hat{C}} < \sin\left(F\gamma_u\right) < \frac{\hat{D} + \pi/2}{\pi\hat{C}}, \\ C_{\mathrm{P}}^{II}\cos^{n_d}\left(\gamma_d + \phi_d\right), & \text{otherwise,} \end{cases} \tag{A5}$$

where $\hat{C} = CE/\Delta X$ and $\hat{D} = D - \pi C\Phi_0$, while $n_d$ and $\phi_d$ are, respectively, the power loss exponent and phase asymmetry of the downstream turbine.

The normalized power at the downstream turbine $j$ affected by the wake released by the upstream turbine $i$ can be written as

$$P_{\mathrm{n},j} = C_{\mathrm{P}}^{II}\left(1 - \delta_i(X_j)A_{i\to j}\right)^3 \cos^{n_j}\left(\gamma_j + \phi_j\right), \tag{A6}$$





where $\delta_i(X_j)$ is the speed deficit of the wake of turbine $i$ at the downstream distance $X_j$ where turbine $j$ is located, and $A_{i \to j}$ is the fractional overlap area of the rotor of $j$ with the wake of $i$. Using Eq. (A5) and Eq. (A6), the speed deficits caused by turbine-to-turbine wake interactions can be readily obtained. In fact, the deficit at turbine $j$ caused by the wake released by turbine $i$ is computed as

$$\delta_i(X_j)A_{i \to j} = \begin{cases} 1 - \left(1 - B_{ij}\left(1 - \sin\left(\pi\hat{C}_{ij}\sin\left(F_i\gamma_i\right) - \hat{D}_{ij}\right)\right)\right)^{1/3}, & \text{if } \frac{\hat{D}_{ij}-3/2\pi}{\pi\hat{C}_{ij}} < \sin\left(F_i\gamma_i\right) < \frac{\hat{D}_{ij}+\pi/2}{\pi\hat{C}_{ij}}, \\ 0, & \text{otherwise}, \end{cases} \quad (A7)$$

where $B_{ij}$, $\hat{C}_{ij}$, $\hat{D}_{ij}$ and $F_j$ are the corresponding tunable parameters.

Finally, the sum of energy deficits method (Renkema, 2007) is used for combining the wakes of the two upstream turbines to get the normalized power of the third machine:

$$P_{\mathrm{n},3} = C_{\mathrm{P}}^{II}\left(1 - \sqrt{\sum_{j=1}^{2}\left(\delta_j(X_3)A_{j \to 3}\right)^2}\right)^3. \quad (A8)$$

*Author contributions.* FC conducted the main research work, developed the Data-Driven model, was responsible for the execution of the wind tunnel tests and post-processed the results. CLB conceived the experiments, developed the theory of the FLORIS-Augm model, contributed to the interpretation of the results and supervised the whole research project. FC and CLB wrote the manuscript. FC and RW conducted the wind tunnel tests. RW was responsible for the operation and maintenance of the scaled turbines, and implemented the turbine and farm controllers and associated software. JS implemented the FLORIS-Augm model and the wind farm controller, participated in the wind tunnel

tests and in the post-processing and analysis of the results. All authors provided important input to this research work through discussions, feedback and by improving the manuscript.

*Competing interests.* The authors declare that they have no conflict of interest.

*Acknowledgements.* This work has been supported by the CL-WINDCON project, which receives funding from the European Union Horizon 2020 research and innovation program under grant agreement No. 727477. The authors acknowledge Stefano Cacciola, Gabriele Campanardi,

Alessandro Croce, Donato Grassi, Luca Riccobene, Paolo Schito, and Alberto Zasso of the Politecnico di Milano for their help in the conduction of the wind tunnel experiments.



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
