# Peer review of "Wind tunnel testing of wake steering with dynamic wind direction changes"

_Wind Energy Science, 2020_

## Referee Comment (RC1) · Anonymous Referee #1 · 17 Jun 2020

Thank you for this paper. The paper presents an important step in wind farm control, the use of controlled wind tunnel experiments to study the design of dynamic wind farm controllers, in this case by studying the behavior of open loop controllers in a dynamic, but controlled environment. Overall, the work is novel and relevant and convincingly presented and a good contribution to the research literature.

Comments:

Good introduction, with good ties to literature

Section 2.3:

$N_t$ and the concept of time speed up, could you explain a little further what this means and how it works?

Section 3.5: For FLORIS you reference a paper, is this version of FLORIS than different than the public version at https://github.com/NREL/floris ?

Fig. 9: Believe the smoothing effect is influenced by the decision to use both positive and negative yaw, uncertainty has a different effect I think when only a single direction is used, which itself could be a means of handling uncertainty, ie using large angles, but only on one side and shift the peak away from the transition. I worry that the small angles of say the sigma=6 deg case end up providing little difference from the range of offsets which occur naturally at a commercial site

P 17: "FLORIS does not include secondary steering", just to be clear though, the public version of FLORIS does include secondary steering. It could be useful to rename the baseline model something other than simply FLORIS as I worry this mis-represents the current state of the code. Perhaps FLORIS-NoSS or something to that effect?

Perhaps drawing on this point, couldn't one say also that beside the inclusion of secondary steering, the differences between the models might be closed by additional tuning? That is, the problem with the first model might not be its simplicity (although agree not including ss is a problem) but that it is not calibrated?

Maybe going even further, you could say, based on Fig 10, prior to implementing at a site, fine calibration and tuning is not very impactful on the prior estimation of gains, but it becomes critical to the realized gains and therefore upon implementation, and not in the estimation phase, a calibration of the model to collected SCADA data is very beneficial. Could this be a general result of the paper?

Fig 16 and Discussion of: Very interesting! In my experience too, we tend to think of the result of uncertainty as leading to "wrong-way steering" and this being the fundamental problem, and so inclusion of uncertainty leads to very conservative yaw offsets. However, one issue is that practical yaw systems can often lead to a tendency to undershoot desired yaw offsets which aren't accounted for in the static model. Your analysis that the effect of uncertainty is dependent on the fidelity of the model is interesting and

nice to see it come out of the data.

Conclusions: – "A robust implementation may lead to power losses in conditions with weak or absent wake interactions. This might suggest the use of wake steering only around conditions where significant wake effects are expected, whereas it should be switched off elsewhere." – I suspect this could be scheduled on wind speed

––––––––––––––––––––––––––––––

---

## Referee Comment (RC2) · Anonymous Referee #2 · 24 Jun 2020

This paper presents results from wind tunnel experiments that assess the performance of open-loop wake steering controllers for a wind farm consisting of 3 scaled wind turbines. I very much enjoyed reading this manuscript: It is well-written and the results are organized and presented well.

A few suggestions for improvement include the following:

1. To me, "normalized power" is equivalent to "power coefficient". I think the authors mean to refer to the "maximal power coefficient" when they refer to "power coefficient".

2. I prefer the descriptor "locally greedy" as a more accurate name for the standard approach "in which each machine works independently from the others to maximize its own power output". The wind farm is not being greedy, but rather the wind turbines are

[Figure]

WESD
being independently, locally greedy.

3. The US National Renewable Energy Laboratory's most recent release of FLORIS does include secondary steering. It may be helpful to more clearly indicate or label that the FLORIS version used in this paper is an older version (such as labeling it as FLORIS-2018?).

4. Page 7 (and possibly elsewhere): I'm not sure that the "time speed-up factor" is the best "name" for eta_t. When I first read the introductory paragraph of Section 2.3, I was confused. A "speed-up factor" suggests that it is greater than 1, but eta_t = 1/80 « 1. Perhaps just call this a "time scaling factor"?

5. Page 22, line 17: It is stated that "in Fig. 16a, the power gains are equal to about 4 – 6% ..." I didn't understand this, since when I look at Fig. 16a, it looks like the power gains are mostly around 10% or even more.

6. I did not completely follow the derivations in Appendix A. What is the difference between $P_{n,d}$ (in A5) and $P_{n,2}$ (in A1)? In (A5), presumably d can be 2 or 3; if (A1) also applies equally well to the 2nd or 3rd turbine, then write (A1) as such as well. And should the denominator on the left-hand side of equation (A1) be $C_P$ (rather than $C_P^{II}$)? Since there are manipulations with inequalities in arriving at the proper range for sin ($F$ gamma_u) on the right hand side of (A5), it should be more explicitly stated that $E > 0$ (and perhaps any other conditions that are assumed).

7. A comment on Wang et al. (2020): I don't think it is good practice to cite a reference that is "in preparation", since the readers cannot yet access this reference. I will ultimately leave it to the editors of WES to determine whether this is allowable or not.

The following are smaller corrections and suggestions for improvement:

I. Page 1, Line 25, insert the word "a" before "coordinated": "... the turbines in a wind farm operate in a coordinated, collaborative fashion."

II. Page 6, Figure 3: the right-most x-axis label in the left plot is very close to the left-

most x-axis label in the middle plot, making it collectively look like the number "1.15". Would it be possible to separate these two plots by another few millimeters?

III. Page 16, lines 9 and 10: No paragraph break is needed here.

IV. Page 21, Figure 13, left plot, suggest moving legend box: It seems there is room in the upper left of the plot to have the legend box there rather than over the bars actually being plotted.

V. Page 22, line 8, add "the" before "farm": "the total power output at the farm level decreases"

VI. Page 25, line 10, suggested re-wording: "A few observations can be made from the results for the shaft DELs."

---

## Author Comment (AC1) · 27 Jul 2020

**Reply to the Reviewers**

We thank the reviewers for their detailed analysis and constructive inputs. A list of point-by-point replies to the reviewers' comments is reported in the following.

**Replies to Review 1**

1. ***[Reviewer]:*** *Thank you for this paper. The paper presents an important step in wind farm control, the use of controlled wind tunnel experiments to study the design of dynamic wind farm controllers, in this case by studying the behavior of open loop controllers in a dynamic, but controlled environment. Overall, the work is novel and relevant and convincingly presented and a good contribution to the research literature.*

   **[Authors]:** We thank the reviewer for the positive feedback.

2. ***[Reviewer]:*** *Nt and the concept of time speed up, could you explain a little further what this means and how it works?.*

   **[Authors]:** Following the reviewer's suggestion, we have expanded the section that discusses the time speed up.

3. ***[Reviewer]:*** *Section 3.5: For FLORIS you reference a paper, is this version of FLORIS than different than the public version at https://github.com/NREL/floris?*

   **[Authors]:** We used the Matlab implementation of FLORIS provided by TU-Delft, and specifically we used the version that could be downloaded on 30.07.2018 from the repository (Doekemeijer and Storm, 2018). The version used in the paper is therefore different from the public version currently available at https://github.com/NREL/floris. We modified the text to better clarify this point.

4. ***[Reviewer]:*** *Fig. 9: Believe the smoothing effect is influenced by the decision to use both positive and negative yaw, uncertainty has a different effect I think when only a single direction is used, which itself could be a means of handling uncertainty, ie using large angles, but only on one side and shift the peak away from the transition. I worry that the small angles of say the $\sigma_\Phi = 6$ deg case end up providing little difference from the range of offsets which occur naturally at a commercial site.*

   **[Authors]:** In this work we tested the effectiveness of the controller for both positive and negative wind directions, thus experimentally exploring a topic that was not addressed even in recent experimental campaigns conducted in the field (Fleming et al., 2017, 2019; Howland et al., 2019). We agree that one might try to use a single-side yaw to shift the peak away from the transition. However, that raises several other issues, and it probably comes at a performance loss. Besides, the experiments were only conducted with positive and negative yaw policies, so this might be an interesting point for a future investigation, but it clearly falls beyond the scope of the present work.

   We do agree with the reviewer that the small yaw setting computed for $\sigma_\Phi = 6$ deg would be comparable to the range of offsets that occur naturally at a commercial site. The control logic of the yaw controllers commonly implemented onboard commercial wind turbines would indeed often prevent the nacelle to properly track a desired small yaw setting, a fact that could reduce the effectiveness of a wake deflection controller. However, in our experiments all nacelle orientations were directly controlled by the central wind farm controller, and not by the local yaw controller. If such a solution were also used at full-scale, we believe that also the yaw setting computed for $\sigma_\Phi = 6$ deg could induce consistent power gains.

5. ***[Reviewer]:*** *P 17: "FLORIS does not include secondary steering", just to be clear though, the public version of FLORIS does include secondary steering. It could be useful to rename the baseline model something other than simply FLORIS as I worry this mis-represents the current state of the code. Perhaps FLORIS-NoSS or something to that effect?*

   **[Authors]:** Although this was already clearly stated, we changed the text in the paper to clarify that we used a version of FLORIS that differs from the latest release.

6. ***[Reviewer]:*** *P 17: Perhaps drawing on this point, couldn't one say also that beside the inclusion of secondary steering, the differences between the models might be closed by additional tuning? That is, the problem with the first model might not be its simplicity (although agree not including ss is a problem) but that it is not calibrated?*

**[Authors]:** The FLORIS baseline model was tuned with dedicated wind tunnel data, as already stated in the manuscript. We changed slightly the text to stress even more clearly that, because of tuning, the parameters used here differ from the ones provided by Bastankhah and Porté-Agel (2016) and Crespo and Hernández (1996).

7. *[Reviewer]: P 17: Maybe going even further, you could say, based on Fig 10, prior to implementing at a site, fine calibration and tuning is not very impactful on the prior estimation of gains, but it becomes critical to the realized gains and therefore upon implementation, and not in the estimation phase, a calibration of the model to?*

    **[Authors]:** Yes, one could base a first implementation on standard values of the model parameters. Then, after enough data has been collected at the site, the model could retuned to ensure a better performance. Indeed we have a paper just published in Wind Energy Science (Schreiber et al., 2020) where we show that a model can be improved from standard operational data. However, this goes a bit beyond the scope of the present paper.

8. *[Reviewer]: Fig 16 and Discussion of: Very interesting! In my experience too, we tend to think of the result of uncertainty as leading to "wrong-way steering" and this being the fundamental problem, and so inclusion of uncertainty leads to very conservative yaw offsets. However, one issue is that practical yaw systems can often lead to a tendency to undershoot desired yaw offsets which aren't accounted for in the static model. Your analysis that the effect of uncertainty is dependent on the fidelity of the model is interesting and nice to see it come out of the data.*

    **[Authors]:** Thanks.

9. *[Reviewer]: Conclusions: – "A robust implementation may lead to power losses in conditions with weak or absent wake interactions, which could be typical also . This might suggest the use of wake steering only around conditions where significant wake effects are expected, whereas it should be switched off elsewhere. For example, wake steering should not be used "– I suspect this could be scheduled on wind speed.*

    **[Authors]:** Certainly, wind farm control in general should be scheduled on wind speed, as already noted in a few places in the text. However, our sentence has a different meaning: irrespectively of the wind speed, conditions characterized by weak wake interactions (due to speed, but also farm layout, orography, etc.) should probably be avoided.

**Replies to Review 2**

1. *[Reviewer]: This paper presents results from wind tunnel experiments that assess the performance of open-loop wake steering controllers for a wind farm consisting of 3 scaled wind turbines. I very much enjoyed reading this manuscript: It is well-written and the results are organized and presented well.*

    **[Authors]:** We thank the reviewer for the positive feedback.

2. *[Reviewer]: To me, "normalized power" is equivalent to "power coefficient". I think the authors mean to refer to the "maximal power coefficient" when they refer to "power coefficient".*

    **[Authors]:** In the paper, we explained the difference between the standard power coefficient and the normalized power (as defined in Eq. (2)) that we used for presenting the experimental results; indeed, we dedicated a whole subsection (3.1.4) to explain the difference between power coefficient and normalized power. We also think that we properly used the term "power coefficient" throughout the text. When necessary, we refereed to the "maximal power coefficient" as $C_{\mathrm{P}}^{II}$, thus implicitly accounting for the fact that wind turbines operate at their maximal power coefficient at wind speeds lower than rated. We slightly modified the text to stress even further the difference between power coefficient and normalized power.

3. *[Reviewer]: I prefer the descriptor "locally greed" as a more accurate name for the standard approach "in which each machine works independently from the others to maximize its own power output". The wind farm is not being greedy, but rather the wind turbines are being independently, locally greedy.*

    **[Authors]:** We agree, and we changed "greedy" into "locally greedy" when first describing this approach.

4. *[Reviewer]: The US National Renewable Energy Laboratory's most recent release of FLORIS does include secondary steering. It may be helpful to more clearly indicate or label that the FLORIS version used in this paper is an older version (such as labeling it as FLORIS-2018?).*

   **[Authors]:** Please see point 3 of the reply to Review 1.

5. *[Reviewer]: Page 7 (and possibly elsewhere): I'm not sure that the "time speed-up factor" is the best "name" for $\eta_t$. When I first read the introductory paragraph of Section 2.3, I was confused. A "speed-up factor" suggests that it is greater than 1, but $\eta_t = 1/80 \ll 1$. Perhaps just call this a "time scaling factor"?.*

   **[Authors]:** The time in the wind tunnel is accelerated, so we think that it is correct to refer to $\eta_t$ as time speed-up factor. We changed the text to better explain the time speed up (see point 2 of the reply to Review 1).

6. *[Reviewer]: Page 22, line 17: It is stated that "in Fig. 16a, the power gains are equal to about $4 - 6\%$ ..." I didn't understand this, since when I look at Fig. 16a, it looks like the power gains are mostly around 10% or even more.*

   **[Authors]:** We believe that Fig. 16a does report gains that agree with the text. The figure also reports, above each column, the power gain change with respect to the FLORIS LUTs with $\sigma_\Phi = 0$ deg case, as also stated in the paper. Probably the reviewer misinterpreted these numbers as gains.

7. *[Reviewer]: I did not completely follow the derivations in Appendix A. What is the difference between $P_{n,d}$ (in A5) and $P_{n,2}$ (in A1)? In (A5), presumably d can be 2 or 3; if (A1) also applies equally well to the 2nd or 3rd turbine, then write (A1) as such as well. And should the denominator on the left-hand side of equation (A1) be $C_P$ (rather than $C_P^{II}$)? Since there are manipulations with inequalities in arriving at the proper range for $\sin(F\gamma)$ on the right hand side of (A5), it should be more explicitly stated that $E > 0$ (and perhaps any other conditions that are assumed).*

   **[Authors]:** Equation A1 can be used to model the normalized power of the downstream turbine in a two-turbines cluster operating at below rated wind speed, when both turbines are aligned with the wind. Since both turbines in the cluster are operating in region II, the denominator on the left-hand side of Eq. (A1) is the region II power coefficient, and therefore $C_P^{II}$. Similarly, Eq. (A5) can be used to model the normalized power of a misaligned downstream turbine in a two-turbines cluster, accounting for the yaw misalignment of the upstream wind turbine. We changed the text in the appendix to better clarify these points; additionally, we specified the sign of the tunable parameters.

8. *[Reviewer]: A comment on Wang et al. (2020): I don't think it is good practice to cite a reference that is "in preparation", since the readers cannot yet access this reference. I will ultimately leave it to the editors of WES to determine whether this is allowable or not.*

   **[Authors]:** In the meanwhile, the paper has been submitted to Wind Energy Science, and its pre-print version should be shortly available online in the "Discussions" version of the journal.

9. *[Reviewer]: II. Page 1, Line 25, insert the word "a" before "coordinated": "...the turbines in a wind farm operate in a coordinated, collaborative fashion."*

   **[Authors]:** We changed the text as suggested.

10. *[Reviewer]: II. Page 6, Figure 3: the right-most x-axis label in the left plot is very close to the leftmost x-axis label in the middle plot, making it collectively look like the number "1.15". Would it be possible to separate these two plots by another few millimeters?*

    **[Authors]:** We changed the figure as suggested.

11. *[Reviewer]: III. Page 16, lines 9 and 10: No paragraph break is needed here.*

    **[Authors]:** We removed the paragraph break.

12. *[Reviewer]: IV. Page 21, Figure 13, left plot, suggest moving legend box: It seems there is room in the upper left of the plot to have the legend box there rather than over the bars actually being plotted.*

    **[Authors]:** We changed the figure as suggested.

13. *[Reviewer]: V. Page 22, line 8, add "the" before "farm": "the total power output at the farm level decreases".*

    **[Authors]:** We changed the text as suggested.

14. *[Reviewer]: VI. Page 25, line 10, suggested re-wording: "A few observations can be made from the results for the shaft DELs".*

    **[Authors]:** We changed the text as suggested.

We have taken the opportunity to make small editorial changes to the text, in order to improve readability. A revised version of the manuscript is attached to the present reply, with the main changes highlighted in red and blue.

**References**

[revised manuscript text omitted]